# Molecularly distinct striatonigral neuron subtypes differentially regulate locomotion

Jie Dong[1], Lupeng Wang [1], Breanna T. Sullivan [1], Lixin Sun[1], Victor M. Martinez Smith [1], Lisa Chang[1], Jinhui Ding[2], Weidong Le [3,4], Charles R. Gerfen [5] & Huaibin Cai [1] ✉

Striatonigral neurons, traditionally known for promoting locomotion, comprise diverse subtypes with distinct transcriptomic profiles. However, their specific contributions to locomotor regulation remain incompletely understood. Using the genetic markers *Kremen1* and *Calb1*, we demonstrate in mouse models that *Kremen1*+ and *Calb1*+ striatonigral neurons exerted opposing effects on locomotion. *Kremen1*+ neurons displayed delayed activation at locomotion onset but exhibited increasing activity during locomotion offset. In contrast, *Calb1*+ neurons showed early activation at locomotion onset and decreasing activity during locomotion offset. Optogenetic activation of *Kremen1*+ neurons suppressed ongoing locomotion, whereas activation of *Calb1*+ neurons promoted locomotion. Activation of *Kremen1*+ neurons induced a greater reduction in dopamine release than *Calb1*+ neurons, followed by a post-stimulation rebound. Conversely, activation of *Calb1*+ neurons triggered an initial increase in dopamine release. Furthermore, genetic knockdown of GABA-B receptor *Gabbr1* in *Aldh1a1*+ nigrostriatal dopaminergic neurons (DANs) reduced DAN inhibition and completely abolished the locomotion-suppressing effect of *Kremen1*+ neurons. Together, these findings reveal a cell type-specific mechanism within striatonigral neuron subtypes: *Calb1*+ neurons promote locomotion, while *Kremen1*+ neurons terminate ongoing movement by inhibiting *Aldh1a1*+ DAN activity via GABBR1 receptors.

Striatal spiny projection neurons (SPNs) constitute approximately 95% of the neuronal population in the dorsal striatum and play a key role in motor learning, decision-making, and regulating voluntary movement[1,2]. These SPNs can be broadly classified into two main subtypes: dopamine receptor D1 (Drd1)-expressing direct-pathway neurons (dSPNs), which project directly to the *globus pallidus internus* (GPi) and *substantia nigra pars reticulata* (SNr); and dopamine receptor D2 (Drd2)-expressing indirect-pathway SPNs (iSPNs), which project to the *globus pallidus externus* (GPe)[1,2]. The dSPNs primarily project to the SNr and are also referred to as striatonigral neurons. It is well recognized that the dSPNs facilitate movement initiation and the iSPNs restrain unwanted movements[2].

SPNs can be further subdivided into two complementary compartments within the dorsal striatum, namely the patch (or striosome) and matrix compartments, which are characterized by distinct neurochemical markers and input-output connectivity[3–6]. Each compartment contains both dSPN and iSPNs. The patch compartment is typically marked by the expression of μ-opiate receptor (MOR1) in clusters of

[1]Transgenic Section, Laboratory of Neurogenetics, National Institute on Aging, National Institutes of Health, Bethesda, MD 20892, USA. [2]Computational Biology Group, Laboratory of Neurogenetics, National Institute on Aging, National Institutes of Health, Bethesda, MD 20892, USA. [3]Clinical Research Center on Neurological Diseases, the First Affiliated Hospital, Dalian Medical University, Dalian, Liaoning 116011, China. [4]Institute of Neurology, Sichuan Academy of Medical Sciences-Sichuan Provincial Hospital, Medical School of University of Electronics & Technology of China, Chengdu, Sichuan 610045, China. [5]Section on Neuroanatomy, National Institute of Mental Health, National Institutes of Health, Bethesda, MD 20892, USA. ✉e-mail: caih@mail.nih.gov

adjacent SPNs in rodents[4,7], whereas the matrix compartment is labeled by Calbindin (CALB1) expression[8]. Advances in single-cell transcriptomics have identified additional genetic markers for dSPNs, iSPNs, and patch- and matrix-associated SPNs[9,10]. Interesting, although patch markers are predominantly expressed in the patch compartment, some are also present among individual or small clusters of SPNs, referred to as "exopatch" neurons, scattered throughout the matrix compartment[10,11]. Furthermore, any given patch marker typically label only a subset of patch-associated SPNs[12]. Previous research has underscored the significance of patch SPNs in modulating mood, decision-making, and reward processing, while matrix SPNs are implicated in action selection[13]. However, the specific roles of molecularly defined patch-associated dSPNs in locomotion control remain poorly understood.

Both patch and matrix dSPNs innervate aldehyde dehydrogenase 1a1-positive (Aldh1a1+) DANs located in the ventral tier of *substantia nigra pars compacta* (SNc)[14], which are particularly vulnerable to degeneration in Parkinson's disease[15]. Notably, a portion of patch-associated dSPN axon terminals bundle up to form distinctive striosome-dendron bouquet structures, intertwining with the dendrites of *Aldh1a1+* DANs that extend into the SNr[16,17]. This anatomical arrangement likely underlies the more potent presynaptic inhibition of DANs by patch-associated dSPNs in contrast to other inhibitory inputs[18]. Moreover, patch-associated dSPNs exert a prolonged inhibitory influence on DAN neuronal activity via GABA-B receptors, crucially regulating the transition from tonic to burst firing in *Aldh1a1+* nigrostriatal DANs, as evidenced by brain slice recordings[16,19]. Despite these insights, the physiological and behavioral implications of this distinctive striatonigral circuit remain to be explored.

One major challenge to study patch-associated SPNs is the difficulty in definitively characterizing and manipulating these neurons, largely due to their less well-defined neurochemical organizations. In this study, we initially identified the *Kremen1* gene as a specific molecular maker for patch-associated SPNs in the dorsal striatum. Subsequently, we generated a line of *Kremen1*[2A-Cre] knock-in (KI) mice to investigate the anatomy of *Kremen1*-positive (*Kremen1+*) dSPN subpopulations and their functional significance in locomotor control compared to the *Calb1*-positive (*Calb1+*) matrix dSPNs.

## Results

### *Kremen1* transcript is enriched in patch compartments in the dorsal striatum

Using published patch reporter *Nr4a1*-eGFP transgenic mice[20], we isolated the eGFP-containing and the adjacent tissues in the dorsal striatum by laser capture microdissection for bulk RNA sequencing **NCBI accession: PRJNA870469**. In comparison to various patch markers examined previously, such as *Oprm1*, *Tshz1*, *Pdyn*, *Nr4a1*, and *Sepw1*, the expression of *Kremen1* mRNA exhibited substantially higher differentials in the patch versus matrix compartments (Fig. 1a, b). Consistently, *Kremen1* was also identified as a patch marker by an independent single-nuclei transcriptomic study[10]. Further co-localization analyses with *Drd1* and *Drd2* using RNAscope in situ hybridization found that approximately 60% of *Kremen1+* SPNs corresponded to dSPNs, while 35% were identified as iSPNs, resembling the ratio of total dSPNs versus iSPNs in the striatum (Fig. 1c–e). Moreover, *Kremen1+* SPNs were predominantly distributed in the dorsal striatum (Fig. 1f). Within the striatum, *Kremen1+* SPNs constituted 12.7% of total SPNs, 52% of SPNs within the patch compartments (defined by clusters of five or more adjacent *Kremen1+* SPNs), and 8.2% of SPNs in the matrix compartments (Fig. 1g). Notably, the density of *Kremen1+* SPNs was substantially higher in patch compartments compared to matrix compartments (Fig. 1h), although the total number of *Kremen1+* SPNs in the patch compartments was slightly lower than that in the matrix compartments (Fig. 1i). These findings highlight *Kremen1* as a useful genetic marker for identifying a distinct subpopulation of SPNs, with a subset forming patch-like clusters specifically in the dorsal striatum.

*Tshz1* is also enriched in patch compartments[9,10]. We further compared the distribution of *Kremen1+* and *Tshz1+* SPNs, especially dSPNs in the dorsal striatum, by RNAScope in situ hybridization. We found that *Kremen1+* SPNs has a larger distribution in the patch than *Tshz1+* SPNs (Supplementary Fig. 1a, b). There is only around 15-20% overlap between these two dSPN subpopulations (Supplementary Fig. 1c, d). These results suggest that *Kremen1+* and *Tshz1+* mark two largely different subtypes of patch SPNs in the dorsal striatum. Together, they accounted for approximately 70% of patch-associated SPNs (Supplementary Fig. 1e).

### *Kremen1*[2A-Cre] knock-in mice are useful for studying patch-associated SPNs

Given the enrichment of *Kremen1* in patch SPNs, we generated a line of *Kremen1*[2A-Cre] knock-in (KI) mice using CRISPR/Cas9-mediated gene editing (Supplementary Fig. 2). By crossing these mice with Ai14 reporter mice[21], which express the red fluorescent protein tdTomato under Cre-dependent regulation, we observed distinct tdTomato signals scattered within the dorsal but not ventral striatum of *Kremen1*[2A-Cre];Ai14 mice (Fig. 2a). These signals co-localized with the widely used patch marker MOR1[4] (Fig. 2a, b). Consistent with the RNAscope analyses (Fig. 1i), the number of tdTomato+ SPNs is comparable in the patch and matrix compartments, with the patch identified by MOR1 immunostaining (Supplementary Fig. 3). Apart from SPNs, tdTomato signals were also detected in pericytes, with no staining detected in the interneurons or glial cells of the dorsal striatum (Supplementary Fig. 4a, b). The *Kremen1+* SPNs were restricted to the dorsal striatum but distributed throughout the entire dorsal striatum along the rostral to caudal and medial to lateral axes (Supplementary Fig. 5a, b). Beyond the dorsal striatum, tdTomato signals were also visible in the hippocampal regions (Fig. 2c, Supplementary Fig. 5b). Notably, the incoming tdTomato-positive dSPN axons exhibited distinct dendron-bouquet like structures[17], forming connections with the dendrites of DANs in SN (Fig. 2d).

To investigate the projection pattern of *Kremen1+* SPNs, we performed stereotaxic injections of adeno-associated viral (AAV) vectors co-expressing Cre-dependent tdTomato and synaptophysin-fused EGFP (sypEGFP) into the dorsal striatum of *Kremen1*[2A-Cre] KI mice (Fig. 2e). Both tdTomato and sypEGFP signals were found within patch-like compartments in dorsal striatum (Fig. 2f). Additionally, sypEGFP specifically labeled the axon terminals of *Kremen1+* SPNs in the GPe and entopeduncular nucleus (EPN), the mouse equivalent to GPi (Fig. 2g), as well as in the SNr and SNc (Fig. 2h). When similar AAV injections were performed in *Calb1*[IRES2-Cre] mice, tdTomato and sypEGFP signals were observe in the matrix compartment, characterized by low MOR1 expression (Supplementary Fig. 6a, b). The axon terminals of *Calb1+* SPNs were found projecting to the GPe, EPN, and SN (Supplementary Fig. 6c, d). In the SNr, a clear overlap was observed between the axon terminals of *Calb1+* dSPNs and the dendrites of TH+ DANs (Supplementary Fig. 6d, e). Therefore, *Kremen1*[2A-Cre] and *Calb1*[IRES2-Cre] KI mice provide useful tools for studying the anatomy and functional roles of patch- or matrix-associated SPNs, respectively.

### Distinct activity changes in *Kremen1+* and *Calb1+* dSPNs during locomotion onset and offset

To investigate the activity patterns of *Kremen1+* and *Calb1+* dSPNs during self-paced locomotion, we injected AAVs expressing the genetically encoded calcium indicator GCaMP8s (AAV1-FLEX-GCaMP8s) into the dorsal striatum and implanted an optic fiber in the SNr of the same hemisphere in *Kremen1*[2A-Cre] or *Calb1*[IRES2-Cre] mice[21] (Fig. 3a, b). The mice were head-fixed and allowed to walk freely on a belt treadmill, where movement velocity signal was synchronized with a fiber photometry setup for recording calcium transients in the axon terminals of *Kremen1+* or *Calb1+* SPNs (Fig. 3c–e, Supplementary Fig. 7). Representative sample traces revealed that calcium transients in the

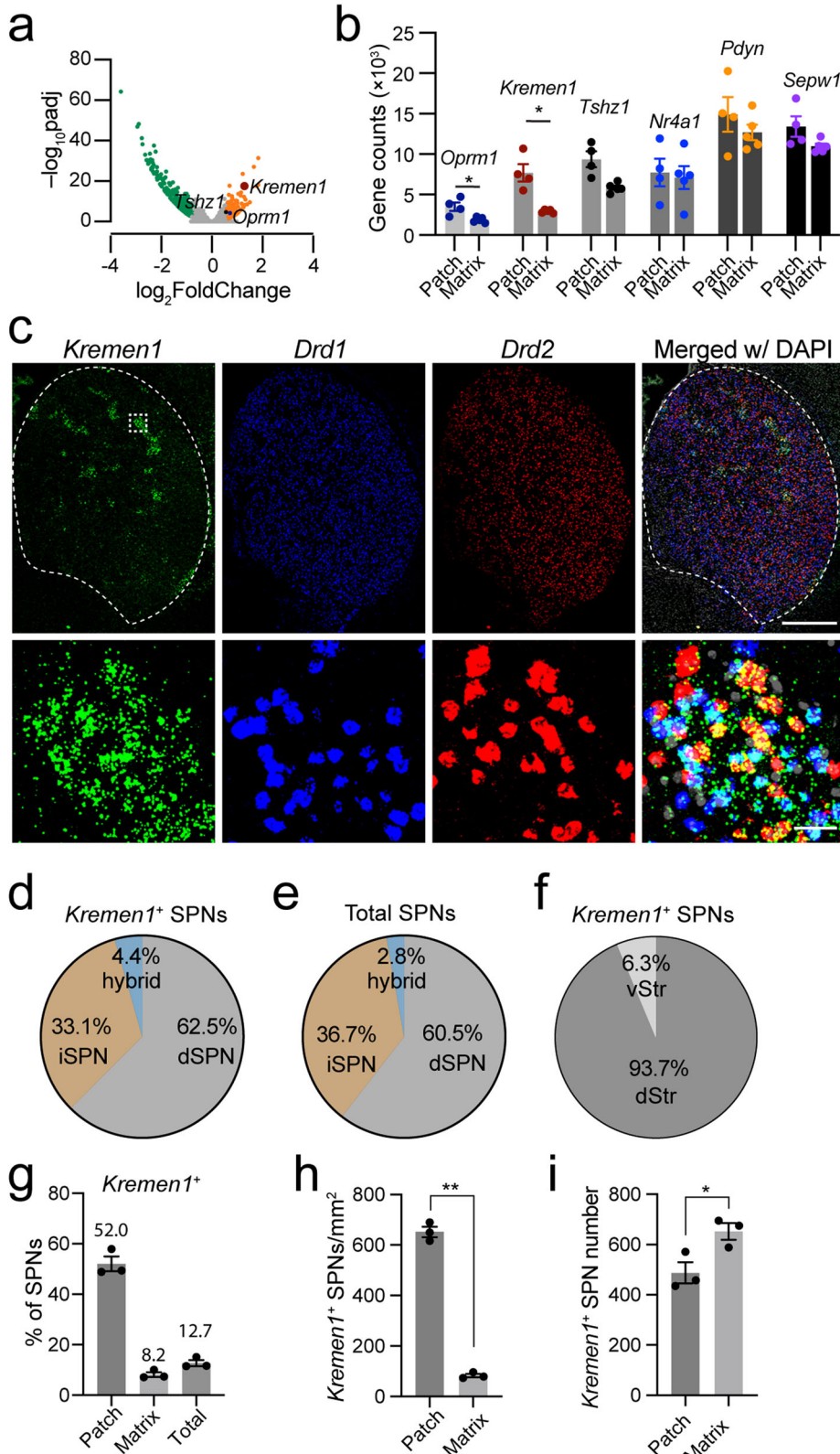

axon terminals of both *Kremen1*[+] and *Calb1*[+] dSPNs correlated with velocity changes during locomotion bouts (Fig. 3f, g). However, distinct differences emerged when aligning their mean activity to locomotion onset and offset (Fig. 3h, i). At locomotion onset, *Kremen1*[+] dSPNs reached their peak activity later compared to *Calb1*[+] dSPNs (Fig. 3j). Notably, *Calb1*[+] dSPN activity began to rise prior to locomotion onset, whereas *Kremen1*[+] dSPN activity increased much later and

mostly after locomotion onset (Fig. 3k). At locomotion offset, the peak activity timing of *Kremen1*[+] dSPNs was delayed compared to *Calb1*[+] dSPNs (Fig. 3l). Furthermore, the slope of activity preceding locomotion offset differed markedly between the two subtypes: *Kremen1*[+] dSPNs exhibited an increasing trend (positive slope), while *Calb1*[+] dSPNs displayed a decreasing trend (negative slope) (Fig. 3m). These distinct activity patterns highlight their differential roles in regulating

**Fig. 1 | Distribution of *Kremen1*⁺ SPNs in patch and matrix compartments.**
**a** Volcano plot shows differential gene expression in patch (*n* = 4 mice) and matrix (*n* = 5 mice) compartments, of which *Kremen1*, *Oprm1*, and *Tshz1* are enriched in patch neurons. **b** Scatter plot compares the expression levels of selective genes in patch (*n* = 4 mice) and matrix (*n* = 5 mice) compartments. Data were presented as mean ± SEM. Paired t-test, two-tailed, *Oprm1*: *p* = 0.0406; *Kremen1*: *p* = 0.0218; *Tshz1*: *p* = 0.0693; *Nr4a1*: *p* = 0.8132; *Pdyn*: *p* = 0.5581; *Sepw1*: *p* = 0.1182. **c** Representative confocal images of RNAscope for *Kremen1*, *Drd1*, *Drd2* and DAPI in the striatum. Bottom, high-magnification images of the boxed area in the top panel. Scale bars: 500 μm (top panel), 20 μm (bottom panel). **d** Percentage of *Drd1*⁺, *Drd2*⁺ and *Drd1/Drd2*⁺ SPNs among *Kremen1*⁺ SPNs in the striatum (*n* = 3 mice). **e** Percentage of *Drd1*⁺, *Drd2*⁺ and *Drd1/Drd2*⁺ SPNs among total SPNs in the striatum (*n* = 3 mice). **f** The percentages of *Kremen1*⁺ SPNs distributed in the dorsal (dStr) and ventral (vStr) striatum (*n* = 3 mice). **g** The percentages of *Kremen1*⁺ SPNs distributed in the patch and matrix compartments, as well as the entire dorsal striatum (*n* = 3 mice). **h** The densities of *Kremen1*⁺ SPNs in the patch compartment (652.2 ± 21.01 cells / mm²) is significantly higher in matrix compartment (83.32 ± 6.94 cells / mm²) across three different mice. Paired t-test, two-tailed, t₍₂₎ = 17.44, **p* = 0.0033. **i** The average number of *Kremen1*⁺ SPNs in the patch compartment (487.9 ± 41.93 cells) and matrix compartment (652.8 ± 32.94 cells) was quantified from five representative coronal hemisphere sections across three different mice. Paired t-test, two-tailed, t₍₂₎ = 6.07, **p* = 0.026. All error bars are represented as mean ± SEM.

motor activity, with *Kremen1*⁺ dSPNs potentially facilitating locomotion termination and *Calb1*⁺ dSPNs contributing to movement initiation. When comparing the locomotion speed of individual mice, the mean velocity at locomotion onset was slightly higher in *Calb1*^IRES2-Cre mice than in *Kremen1*^2A-Cre mice (Supplementary Fig. 8a–c). However, no significant differences in velocity were observed at locomotion offset, nor were there differences in the shape of velocity changes between the two groups at either locomotion onset or offset (Supplementary Fig. 8a, b, d). Nonetheless, we do not believe that mouse strain differences account for the distinct dynamic changes in *Kremen1*⁺ and *Calb1*⁺ dSPN activity during locomotion onset and offset.

### *Kremen1*⁺ and *Calb1*⁺ dSPNs exert contrasting roles in locomotor control

To examine the causal influence of *Kremen1*⁺ dSPN activity on locomotion, we used optogenetic manipulations by introducing AAV vectors containing Cre-dependent optogenetic activator Channelrhodopsin-2 (AAV1-FLEX-ChR2) into the dorsal striatum of *Kremen1*^2A-Cre mice and implanted the optic fiber in the SN for light stimulation of *Kremen1*⁺ dSPN axon terminals (Fig. 4a, b). Optogenetic activation at 3 mW/20 Hz for 3 min led to a notable reduction in locomotion velocity, followed by a subsequent rebound in motor activity after the stimulation period (Fig. 4c, d). The decrease in walking speed likely resulted from both a reduction in velocity during each bout of ambulatory movement (Fig. 4e) and reduced locomotor activity, as evidenced by reduced movement frequency and shortened duration (Fig. 4f, g), as well as an increase in immobility frequency (Fig. 4h). This reduction in locomotion velocity was also observed at various stimulation parameters, including continuous 0.25 mW power intensities, 10 Hz stimulation frequency, and 10 s durations (Supplementary Fig. 9a–c). In contrast, optogenetic activation of *Calb1*⁺ dSPNs of *Calb1*^IRES2-Cre mice yielded opposite outcomes, resulting in a substantial increase in movement velocity, coupled with an increase of ambulatory speed and duration, as well as a decrease in immobility frequency and duration (Fig. 4i–p, Supplementary Fig. 9d–f). Taken collectively, these findings demonstrate that *Kremen1*⁺ and *Calb1*⁺ dSPNs exert contrasting roles in locomotor control.

### State-dependent regulation of locomotion by *Kremen1*⁺ and *Calb1*⁺ dSPNs

To examine whether the effects of *Kremen1*⁺ and *Calb1*⁺ dSPN activation depend on locomotion states, we further analyzed the 10 s 3 mW/20 Hz stimulation dataset (Supplementary Fig. 9c, f), categorizing trials into quiescent or ambulatory states immediately preceding stimulation onset or after offset (Fig. 5a–d). Activation of *Kremen1*⁺ dSPNs immediately suppressed ongoing ambulatory movement but slightly increased movement when mice were in a quiescent state (Fig. 5a, c). Conversely, activation of *Calb1*⁺ dSPNs had no immediate effect on ongoing ambulatory movement but triggered ambulatory locomotion when mice were in a quiescent state prior to stimulation (Fig. 5b, d). Additionally, the termination of *Kremen1*⁺ dSPN activation led to increased locomotion velocity, while the termination of *Calb1*⁺ dSPN stimulation resulted in reduced movement speed (Fig. 5e, f).

Therefore, these findings highlight the opposing roles of *Kremen1*⁺ and *Calb1*⁺ dSPNs in locomotion regulation, which are influenced by the locomotion state at the time of activation.

### *Kremen1*⁺ iSPNs play limited role in locomotor control

To investigate the role of *Kremen1*⁺ iSPNs in the context of locomotion, we introduced AAV vectors containing optogenetic activators that were both Cre and Flp-dependent (AAV1-ConFon-ChR2) into the dorsal striatum of *Kremen1*^2A-Cre;*A2a*^2A-Flp double KI mice (Fig. 6a). *Kremen1*⁺ iSPNs project exclusively to the GPe and not to the EPN or SNr (Fig. 6b, c; Supplementary Fig. 10a). Light stimulation was applied at the axon terminals of *Kremen1*⁺ iSPNs situated in the GPe (Fig. 6b, c; Supplementary Fig. 10b). The high-power stimulation at 3 mW/20 Hz for 3 min resulted in a modest increase in average velocity and duration per bout of movement (Fig. 6d–h), while at low light power level with 0.25 mW constant stimulation we did not observe any notable alteration in locomotion velocity (Fig. 6i). These results underscore the limited role of *Kremen1*⁺ iSPNs in regulating locomotion. Indeed, chemogenetic activation of both *Kremen1*⁺ dSPNs and iSPNs in *Kremen1*^2A-Cre mice led to reduced locomotor activity (Supplementary Fig. 11), supporting a major role of *Kremen1*⁺ dSPNs in suppressing locomotion.

### *Kremen1*⁺ and *Calb1*⁺ dSPNs differently regulate dopamine release

To investigate how *Kremen1*⁺ and *Calb1*⁺ dSPNs distinctly regulate locomotion at the circuit level, we examined the impact of dSPN activity on dopamine release in the dorsal striatum. Both *Kremen1*⁺ and *Calb1*⁺ dSPNs innervate nigrostriatal DANs and modulate their activity[14,16,18,22]. Previous studies have shown that optogenetic stimulation of patch neurons can inhibit dopamine release[23,24]. Using the genetically encoded dopamine sensor GRAB_rDA3m[25] and Cre-dependent ChR2, we assessed dopamine release in the dorsal striatum of *Kremen1*^2A-Cre (Fig. 7a, b) and *Calb1*^IRES2-Cre (Fig. 7a, c) mice through fiber photometry following optogenetic stimulation at the SNr for each mouse line. Activation of *Kremen1*⁺ dSPNs led to a gradual reduction in dopamine levels in a stimulation dose dependent manner, with stimulation frequency of 20 Hz and durations of 2, 5, and 15 seconds (Fig. 7d). In contrast, activation of *Calb1*⁺ dSPNs induced a biphasic change in dopamine release: an initial transient increase in dopamine signals at the onset of optic stimulation, followed by a subsequent reduction during the stimulation (Fig. 7d). The initial surge of dopamine release following *Calb1*⁺ dSPN activation was present across different stimulation frequencies and durations (Fig. 7e). However, the magnitude of dopamine release reduction was far less pronounced with *Calb1*⁺ dSPNs stimulation compared to *Kremen1*⁺ dSPNs (Fig. 7f). Additionally, only *Kremen1*⁺ dSPN stimulation resulted in a post-stimulation rebound of dopamine release (Fig. 7g), consistent with a previous observation of DANs rebound firing following patch dSPN stimulation in brain slice[16]. The same pattern of differential dopamine release regulation by *Kremen1*⁺ and *Calb1*⁺ dSPNs was also observed when stimulating the neurons at 10 Hz (Supplementary Fig. 12). These findings suggest that *Kremen1*⁺ and *Calb1*⁺ dSPNs

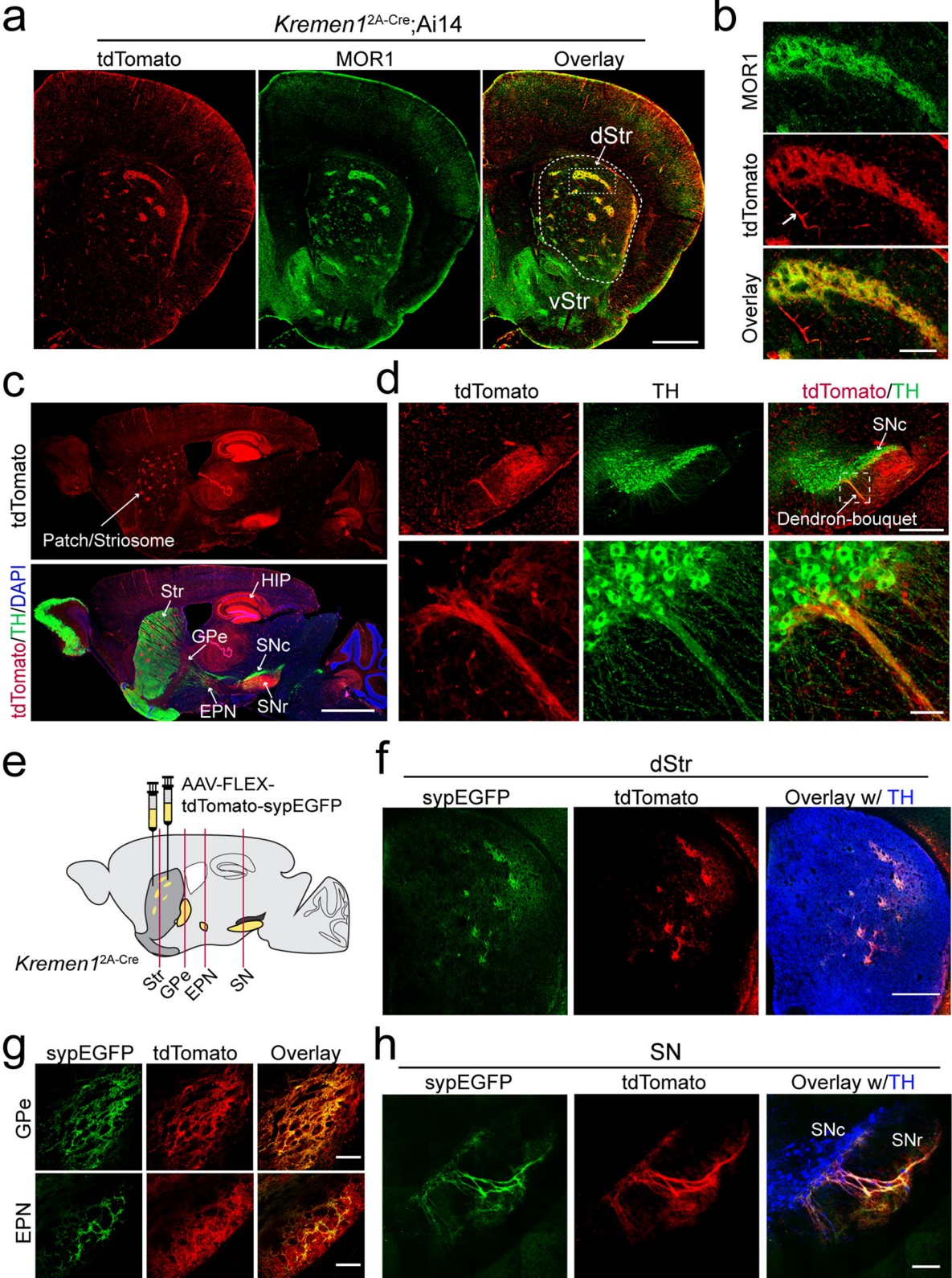

regulate dopamine release differently, with *Kremen1*⁺ dSPN activation leading to a more potent inhibition of dopamine release.

### Genetic knockdown of GABA-B receptor *Gabbr1* in *Aldh1a1*⁺ DANs promotes locomotion

Considering that *Aldh1a1*⁺ nigrostriatal DANs are major recipients of striatal inputs in the SNc and receive prolonged inhibitory signals mediated by GABA-B receptors from patch dSPNs[16,19] and that GABBR1 is the primary GABA receptor in DANs (plotted from **PRJNA775656**[26], Supplementary Fig. 13a), we genetically deleted *Gabbr1* in *Aldh1a1*⁺ nigrostriatal DANs. This was achieved through the delivery of Cre-dependent CRISPR/saCas9-*Gabbr1*sgRNA targeting AAVs into the SNc of *Aldh1a1*CreERT2 mice[14] (Fig. 8a). The gene targeting-vector specifically disrupted the expression of GABBR1 in *Aldh1a1*⁺ DANs, while sparing

**Fig. 2 | Distinct distribution patterns of *Kremen1*+ SPNs in the dorsal striatum and their projections. a** Representative images of tdTomato (red) MOR1 (green) in the dorsal striatum of *Kremen1*2A-Cre;Ai14 mice. dStr: dorsal *s*triatum. vStr: ventral striatum. Scale bar: 1 mm. **b** Enlarged images of the boxed region in (**a**). Arrow points to pericytes that also express tdTomato. Scale bar: 200 μm. **c** Sample images of a sagittal brain section from a representative *Kremen1*2A-Cre;Ai14 mouse. The tdTomato signals are prominent in the patch compartments of Str, GPe, EPN, SN and hippocampus (HIP). Scale bar: 2 mm. **d** Representative images of the SN area, showing the connection of tdTomato-positive SPN axon terminals with the

dendrites of dopaminergic neurons in the SNr, forming the so-called dendron-bouquet structure. Bottom panel shows the magnification of boxed area from the top. Scale bars: 500 μm (top panel), 50 μm (bottom panel). **e** Schematics of AAV9-FLEX-tdTomato-synEGFP injection in the dorsal striatum of *Kremen1*2A-Cre mice. The red vertical lines indicate the location of coronal sections in (**f**–**h**). **f**–**h** Images of tdTomato (red), sypEGFP (green) and TH (blue) staining in dorsal striatum (dStr), *globus pallidus externus* (GPe), entopeduncular nucleus (EPN), and *substantia nigra* (SN). Scale bars: 500 μm in dStr, 200 μm in GPe and EPN, 100 μm in SN.

*Aldh1a1*-negative neurons (Fig. 8b–e, Supplementary Fig. 14). The resulting knockdown (KD) of *Gabbr1* in *Aldh1a1*+ nigrostriatal DANs led to a significant increase in ambulatory velocity (Fig. 8f, g). These findings support an important role of GABA-B receptor-mediated inhibitory signaling in *Aldh1a1*+ DANs for regulating locomotor control.

### *Kremen1*+ dSPNs suppress locomotion through the GABA-B receptors in *Aldh1a1*+ DANs

To investigate the role of postsynaptic GABA-B receptors in the locomotor inhibition induced by *Kremen1*+ dSPNs, we selectively activated *Kremen1*+ dSPNs through optogenetics while simultaneously downregulating *Gabbr1* in *Aldh1a1*+ nigrostriatal DANs of *Kremen1*2A-Cre;*Aldh1a1*CreERT2 double KI mice (Fig. 9a, b). Notably, the only *Kremen1*+ cells present in the SN are pericytes, which do not express GABBR1 (Supplementary Fig. 13b). Since baseline locomotion differed between *Gabbr1*-KD and control mice (Fig. 8f, g), we computed a velocity ratio normalized to each mouse's baseline for group comparison. Remarkably, *Gabbr1* KD in *Aldh1a1*+ nigrostriatal DANs completely abolished both the locomotion-suppressing effects during light stimulation and the post-stimulation rebound typically induced by *Kremen1*+ dSPN activation (Fig. 9c–e). In contrast, control mice showed a marked reduction in velocity during stimulation and a rebound effect after stimulation (Fig. 9c–e).

To access whether GABA-B receptors are involved in locomotor regulation by *Kremen1*-negative dSPNs, we optogenetically stimulated *Kremen1*-negative dSPNs in *Kremen1*2A-Cre;*Aldh1a1*CreERT2 double KI mice under both control and *Gabbr1*-KD conditions (Fig. 9f, g; Supplementary Fig. 15). While dSPN stimulation substantially increased locomotion in both groups, the absence of GABBR1 in *Aldh1a1*+ nigrostriatal DANs did not alter the locomotor effects induced by *Kremen1*-negative dSPNs (Fig. 9h–j). Together, these findings demonstrate that *Kremen1*+ dSPNs suppress locomotor activity by modulating *Aldh1a1*+ nigrostriatal DAN activity via GABBR1 signaling.

### *Kremen1*+ dSPNs regulate *Aldh1a1*+ DAN activity via GABBR1 receptors

To further confirm that *Kremen1*+ dSPNs mediate locomotor regulation by modulating *Aldh1a1*+ DAN activity through GABBR1 receptors, we used fiber photometry to selectively measure calcium transients in the axon terminals of *Aldh1a1*+ DANs during optical activation of *Kremen1*+ dSPNs under both control and *Gabbr1* KD conditions (Fig. 10a, b). When stimulated at 20 Hz for 15 s, optical activation of *Kremen1*+ dSPNs resulted in a significantly smaller reduction in calcium transients in the axon terminals of *Aldh1a1*+ DANs in *Gabbr1* KD mice compared to controls (Fig. 10c, d; **individual traces**, Supplementary Fig. 16). Additionally, under the same stimulation conditions, *Gabbr1* KD abolished the rebound activity of *Aldh1a1*+ DANs following *Kremne1*+ dSPN activation (Fig. 10c, **e**. These findings underscore a critical role of *Aldh1a1*+ DANs in *Kremen1*+ dSPN-regulated locomotor control and establish GABBR1 receptors as an important mediator (Fig. 10f).

## Discussion

In this study, we utilized the distinct expression patterns of *Kremen1* and *Calb1* in dorsal striatum to investigate the functional roles of diverse dSPN subtypes in locomotor controls. We found that while

the activation of *Calb1*+ dSPNs promoted locomotion, consistent with the traditional role of dSPNs, *Kremen1*+ dSPNs– a subset of which form patch-like clusters in the dorsal striatum– suppressed locomotion. This regulation of locomotion by *Kremen1*+ and *Calb1*+ dSPNs is state-dependent, varying with the locomotor context during their activation. Although dSPNs are traditionally known to promote locomotion by inhibiting inhibitory outputs from the SNr[1], our findings reveal that *Kremen1*+ dSPNs suppress locomotion through inhibition of *Aldh1a1*+ DANs via GABBR1-mediated signaling. These results underscore the complex regulation of locomotion by distinct dSPN subpopulations and highlight the value of molecular makers, combined with targeted neuronal manipulation and recording, in elucidating the specialized functions of different neuronal subtypes and circuits.

While prior studies suggest that patch-like dSPNs constrain locomotion[24,27–29], these findings do not necessarily generalize to the entire patch dSPN population. It is important to note that defining "patch" structures based on marker expression can be somewhat arbitrary. Here, we defined putative patch structures as groups of five or more adjacent *Kremen1*+ SPNs and manually delineated these clusters, which include both *Kremen1*+ and *Kremen1*-negative neurons. Approximately 70% of these circled neurons were either *Kremen1*+ or *Tshz1*+. Notably, patch markers are also expressed by a considerable number of neurons within matrix compartments[10,11,20]. Although the density of *Kremen1*+ SPNs is significantly lower in matrix compartments than in patch compartments, the total number of *Kremen1*+ SPNs in the two compartments is comparable. Given this distribution, we cannot distinguish whether *Kremen1*+ dSPN axon terminals originate from patch or matrix compartments under the current experimental conditions. Therefore, we avoided using the terms "patch" or "matrix" to describe *Kremen1*+ and *Calb1*+ SPNs, referring to them instead as molecularly defined SPN subpopulations.

The role of GPi-projecting *Tshz1*+ dSPNs has also been studied in locomotion, with *Tshz1*+ patch dSPNs implicated in locomotor suppression during place-preference tasks[28]. However, changes in movement velocity during valence contexts (e.g., place preference tests) may reflect alterations in active avoidance or approach choices rather than pure locomotion[13]. Moreover, SNr-projecting dSPNs, which are known for locomotion control, were not explicitly examined in prior studies[28]. Our data reveal that *Kremen1* and *Tshz1* dSPNs represent largely different subtypes of patch-like dSPNs. While *Kremen1*+ dSPNs inhibit locomotion via SNr projections, the role of SNr-projecting *Tshz1*+ SPNs remains unclear. These findings emphasize the need for future studies to explicitly identify neuronal subtypes based on both molecular markers and projection targets.

*Kremen1*+ dSPN activity increases much later than *Calb1*+ dSPNs and mostly after locomotion onset, suggesting they do not initiate movement but instead modulate or terminate it. In contrast, *Calb1*+ dSPNs show elevated activity that precedes locomotion onset, consistent with a role in movement initiation. At the locomotion offset, *Kremen1*+ dSPNs exhibit elevated and upward-trending neuronal activity compared to *Calb1*+ dSPNs, suggesting that they play a role in terminating movement and facilitating state transitions from locomotion to quiescence. This aligns with the established principle of basal ganglia involvement in state transition regulation[30]. Optogenetic

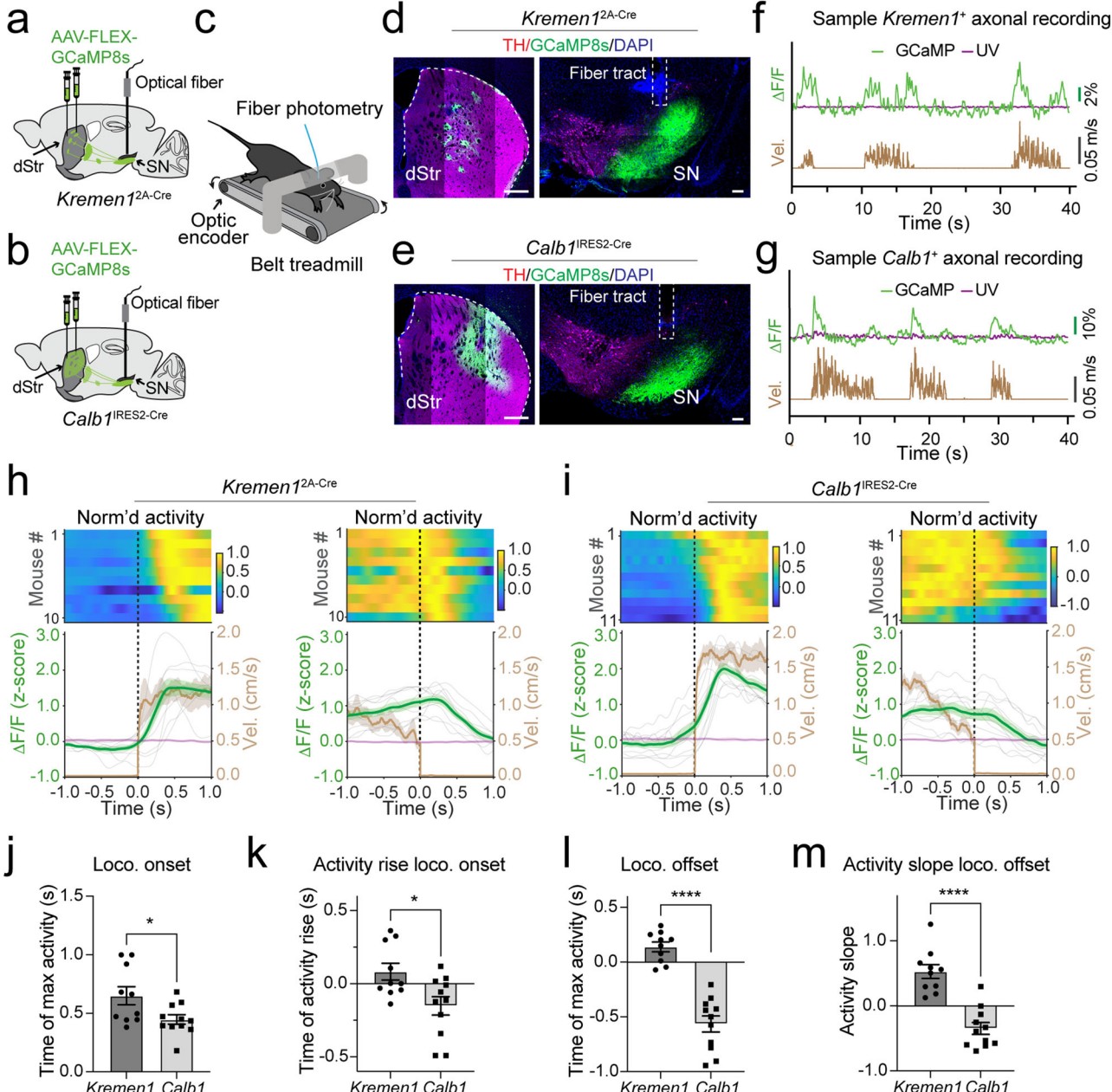

**Fig. 3 | Activity of *Kremen1*+ and *Calb1*+ dSPNs during self-paced locomotion. a, b** Schematics outlines AAV injection and optic fiber implantation for recording the activity of *Kremen1*+ (**a**) and *Calb1*+ dSPNs (**b**). **c** Cartoon shows a head-fixed mouse with optic fiber implantation on a belt treadmill. **d, e** Images of GCaMP8s (green), TH (magenta), and DAPI (blue) in the dStr and SN of *Kremen1*2A-Cre (**d**) and *Calb1*IRES2-Cre mice (**e**). Fiber track is indicated in the SN. Scale bar: 500 μm (dSTR) and 100 μm (SN). **f, g** Sample traces of GCaMP8s (green) and UV isosbestic (purple) with corresponding velocity (brown) traces from a representative *Kremen1*2A-Cre (**f**) and a *Calb1*IRES2-Cre mouse (**g**). **h, i** The top heatmaps show normalized mean activity of *Kremen1*+ (**h**) and *Calb1*+ (**i**) dSPNs for individual mice. The bottom graphs depict population mean (green) and individual animal data (gray) for *Kremen1*+ (**h**, *n* = 10 mice) and *Calb1*+ (**i**, *n* = 11 mice) dSPN activity, along with population mean UV isosbestic (purple) and population mean velocity traces (brown), aligned to locomotion onset (left) and offset (right). The data are presented as mean (solid line) ± SEM (shaded area). **j** Timing of maximum activity during locomotion onset. *Kremen1* (*n* = 10 mice): 0.65 ± 0.08 s; *Calb1* (*n* = 11 mice): 0.45 ± 0.04 s. Unpaired t-test, two-tailed, t(19) = 2.40, *p = 0.027. **k** Timing of activity rose above baseline during locomotion onset. *Kremen1* (*n* = 10 mice): 0.08 ± 0.06 s; *Calb1* (*n* = 11 mice): -0.15 ± 0.06 s. Unpaired t-test, two-tailed, t(19) = 2.73, *p = 0.013. **l** Timing of maximum activity during locomotion offset. *Kremen1* (*n* = 10 mice): 0.14 ± 0.04 s; *Calb1* (*n* = 11 mice): -0.56 ± 0.07 s. Unpaired t-test, two-tailed, t(19) = 8.01, ****p = 1.6e-7. **m** Slope of activity preceding locomotion offset. *Kremen1* (*n* = 10 mice): 0.53 ± 0.11; *Calb1* (*n* = 11 mice): -0.35 ± 0.09. Unpaired t-test, two-tailed, t(19) = 6.27, ****p = 5.1e-6. All error bars are presented as mean ± SEM.

activation of *Kremen1*+ dSPNs shortened movement bout duration and increased immobility, supporting their role in transitioning between locomotor states.

*Aldh1a1*+ nigrostriatal DANs are the primary recipients of monosynaptic inputs from the dorsal striatum and form reciprocal innervation with dSPNs[14]. This feedback loop likely facilitates the regulation of dopamine release and SPN activity during motor control and learning. Both patch and matrix dSPNs innervate *Aldh1a1*+ DANs[14]. Notably, *Kremen1*+ and *Calb1*+ dSPN axon terminals occasionally form striosome-dendron bouquet-like structures[16,17] with the DAN dendrites in the SNr, a configuration that likely influences dopamine release in the dorsal striatum and SNr[16,18]. Consistent with these in vitro studies, our in vivo recordings reveal a more pronounced inhibition of dopamine release upon stimulating *Kremen1*+ dSPNs compared to the *Calb1*+

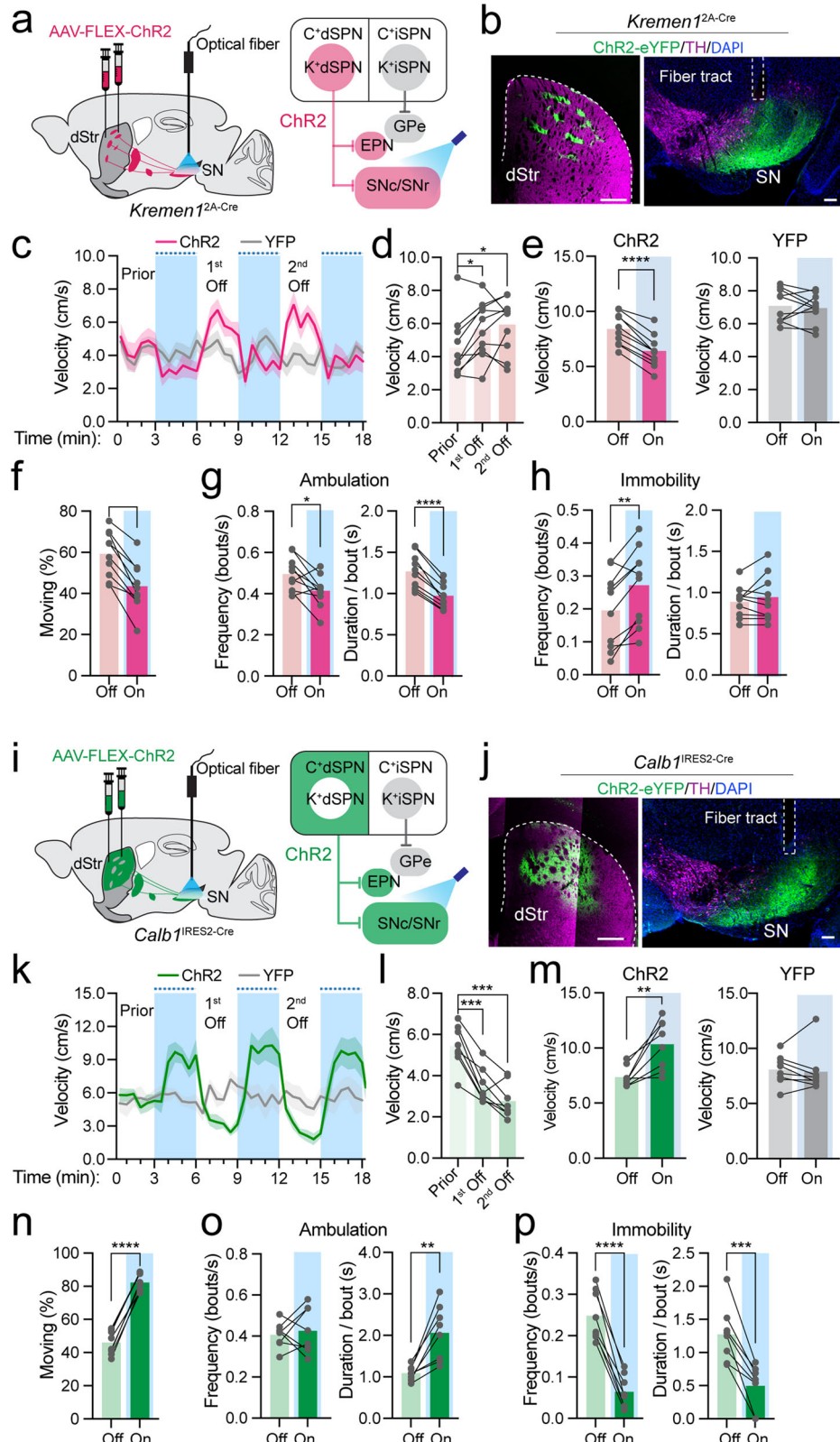

dSPNs, with a rebound dopamine release observed upon the cessation of *Kremen1*+ dSPN stimulation.

Activation of *Calb1*+ dSPNs initially increases dopamine release, likely due to disinhibition of DANs through inhibition of parvalbumin neurons (PVNs) in the SNr, which tonically inhibit DANs[31–34]. Conversely, stimulation of *Kremen1*+ dSPNs results in greater dopamine suppression, followed by rebound release after stimulation cessation.

These opposing effects on dopamine release align with their respective locomotor functions: *Calb1*+ dSPNs promote movement, while *Kremen1*+ dSPNs terminate it. The genetic knockdown of GABBR1 in *Aldh1a1*+ DANs abolished the locomotor suppression and post-stimulation rebound activity induced by *Kremen1*+ dSPNs, confirming the critical role of GABBR1-mediated signaling. Fiber photometry further showed that *Gabbr1* KD in *Aldh1a1*+ DANs reduced the suppression

**Fig. 4 | Optogenetics activation of *Kremen1*⁺ and *Calb1*⁺ dSPNs during locomotion. a, i** Experimental design for activation of *Kremen1*⁺ (**a**) and *Calb1*⁺ (**i**) dSPNs. **b, j** Representative images showing ChR2 (green), TH (magenta) and DAPI (blue) in *Kremen1*²ᴬ⁻ᶜʳᵉ mice (**b**) and *Calb1*ᴵᴿᴱˢ²⁻ᶜʳᵉ mice (**j**). Scale bars: 500 μm (dStr) and 100 μm (SN). **c, k** Instantaneous locomotion velocity of *Kremen1*²ᴬ⁻ᶜʳᵉ (**c**) and *Calb1*ᴵᴿᴱˢ²⁻ᶜʳᵉ mice (**k**) in open-field test. Blue shaded area represents light-on condition. YFP represents yellow fluorescent protein control. The data are presented as mean (solid line) ± SEM (shaded area). **d, l** Comparison of average velocity between prior, 1ˢᵗ Off, and 2ⁿᵈ Off in *Kremen1*-ChR2 (**d**) and *Calb1*-ChR2 mice (**l**). For *Kremen1*-ChR2, one-way ANOVA, F (1.43,12.85) = 5.587, *p = 0.025, multiple comparisons: prior vs. 1ˢᵗ Off, *p = 0.013; prior vs. 2ⁿᵈ Off, *p = 0.035. For *Calb1*-ChR2: F(1.62, 11.31) = 38.96, ***p = 6.3e-5, multiple comparisons: prior vs. 1ˢᵗ Off, ***p = 0.0004; prior vs. 2nd Off, ***p = 0.0004. **e** Average velocity during ambulation bouts in

*Kremen1*-ChR2 ($t_{(9)}$ = 10.08, ****p = 1.9e-6) and *Kremen1*-YFP ($t_{(9)}$ = 0.50, p = 0.63). **f–h** Percentage of ambulation time (**f**), t(9) = 7.41, ****p = 4.1e-5, frequency ($t_{(9)}$ = 2.86, **p = 0.02) and duration ($t_{(9)}$ = 9.67, ****p = 4.8e-5) of ambulation bouts (**g**), frequency ($t_{(9)}$ = 4.4, **p = 0.002) and duration ($t_{(9)}$ = 0.87, p = 0.41) of immobility bouts (**h**) in *Kremen1*-ChR2 mice. n = 10 mice per each group in (**c–h**). **m** Average ambulation velocity in *Calb1*-ChR2 ($t_{(7)}$ = 4.18, **p = 0.004) and *Calb1*-YFP ($t_{(7)}$ = 0.44, p = 0.67). **n–p** Percentage of ambulation time (**n**, $t_{(7)}$ = 22.74, ****p = 8.06e-8), frequency ($t_{(7)}$ = 0.48, p = 0.65) and duration ($t_{(7)}$ = 4.8, **p = 0.002) of ambulation bouts **o**, frequency ($t_{(7)}$ = 11.41, ****p = 8.9e-6) and duration ($t_{(7)}$ = 6.4, ***p = 0.0004) of immobility bouts **p** in *Calb1*-ChR2 mice. n = 8 mice per group in (**k–p**). Comparison between two specific groups was performed using paired t test, two tailed. All error bars represent mean ± SEM.

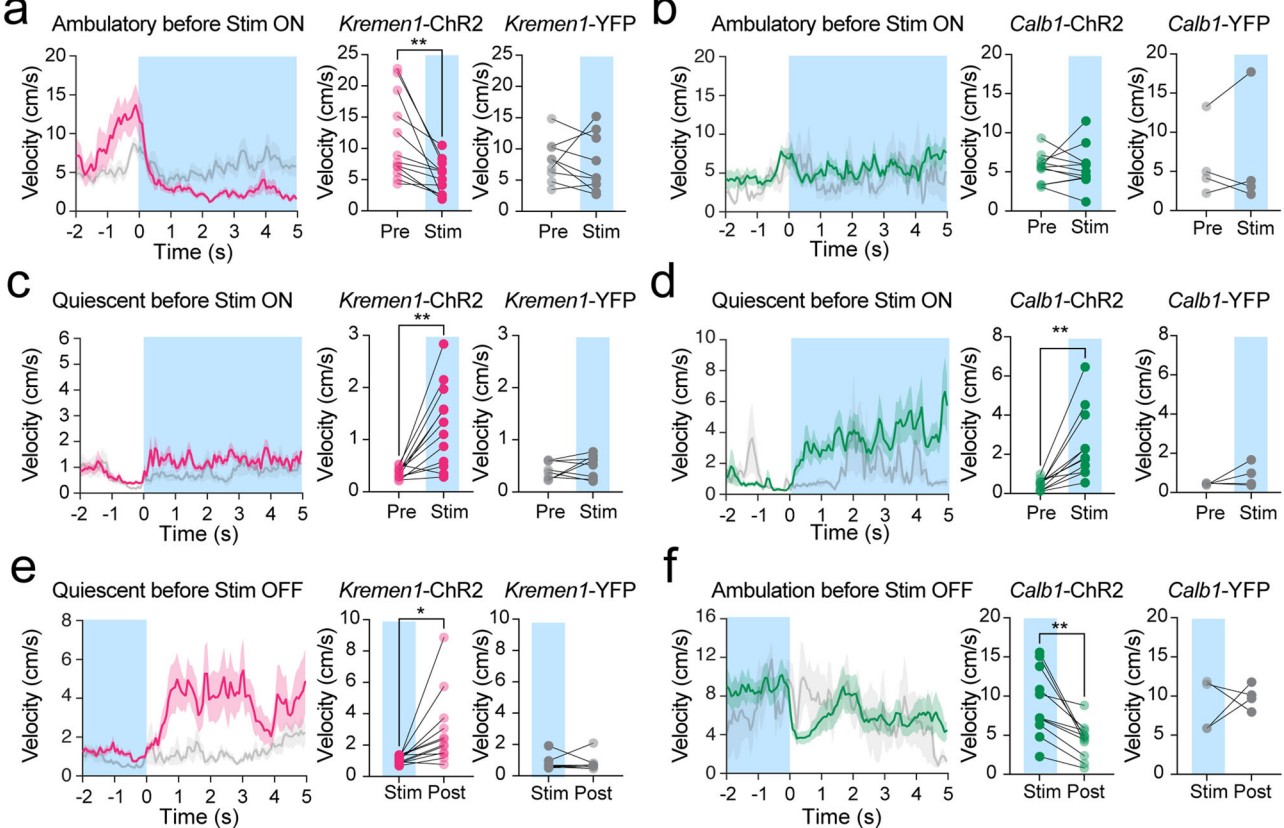

**Fig. 5 | Optogenetics activation *Kremen1*⁺ and *Calb1*⁺ dSPNs exerted state-dependent regulation of locomotion. a, b** Average velocity trace with *Kremen1*⁺ dSPN (**a**, left) and *Calb1*⁺ dSPNs (**b**, left) stimulation when mice were in ambulatory state before stimulation onset, blue area indicates first 5 s stimulation, red/green trace shows ChR2 mice and gray trace shows YFP mice. Mean velocity 1 s before and 1 s in stimulation were used for paired t test, two tailed. *Kremen1*-ChR2 mice (**a**, medial): $t_{(11)}$ = 3.70; **p = 0.0035, n = 12 mice. *Kremen1*-YFP (**a**, right): p = 0.72, n = 9 mice. *Calb1*-ChR2 mice (**b**, medial): $t_{(10)}$ = 0.05, p = 0.96, n = 11 mice. *Calb1*-YFP (**b**, right): p = 0.77, n = 4 mice. **c, d** Average velocity trace with *Kremen1*⁺ dSPN (**c**, left) and *Calb1*⁺ dSPNs (**d**, left) stimulation when mice were in quiescent/low movement state before stimulation onset. Mean velocity 1 s before and 1 s in stimulation were used for paired t-test, two tailed. *Kremen1*-ChR2 mice (**c**, medial):

$t_{(11)}$ = 4.10, **p = 0.0018, n = 12 mice. *Kremen1*-YFP (**c**, right): p = 0.32, n = 9 mice. *Calb1*-ChR2 mice (**d**, medial): $t_{(10)}$ = 4.10, **p = 0.0022, n = 11 mice. *Calb1*-YFP (**d**, right): p = 0.27, n = 4 mice. **e** Average velocity trace with *Kremen1*⁺ dSPN stimulation when mice were in quiescent/low movement state before stimulation offset. Shaded area indicates the last 2 s of stimulation. Mean velocity 1 s before and 1 s post stimulation offset were used for paired t-test, two tailed. *Kremen1*-ChR2 mice: $t_{(11)}$ = 2.86, *p = 0.016, n = 12 mice. *Kremen1*-YFP: p = 0.91, n = 9 mice. **f** Average velocity trace with *Calb1*⁺ dSPN stimulation when mice were in ambulatory state before stimulation offset. Mean velocity 1 s before and 1 s post stimulation offset were used for paired t-test, two tailed. *Calb1*-ChR2 mice: $t_{(11)}$ = 4.26 **p = 0.0017, n = 11 mice. *Calb1*-YFP: p = 0.66, n = 4 mice. All error bars represent mean ± SEM. The data are presented as mean (solid line) ± SEM (shaded area) in velocity traces.

of DAN activity and abolished the rebound response following *Kremen1*⁺ dSPN activation.

Our study demonstrates that molecularly distinct dSPN subpopulations play specialized roles in locomotion control. *Calb1*⁺ dSPNs promote movement by suppressing inhibitory outputs from SNr, whereas *Kremen1*⁺ dSPNs primarily inhibit locomotion by suppressing *Aldh1a1*⁺ DANs through GABBR1-mediated signaling. These findings emphasize the importance of molecular markers in

dissecting neuronal circuit mechanisms and highlight the intricate interplay between distinct dSPN subpopulations in basal ganglia motor control.

## Methods
### Mouse work
All mouse studies were conducted in accordance with the guidelines approved by the Institutional Animal Care and Use Committees

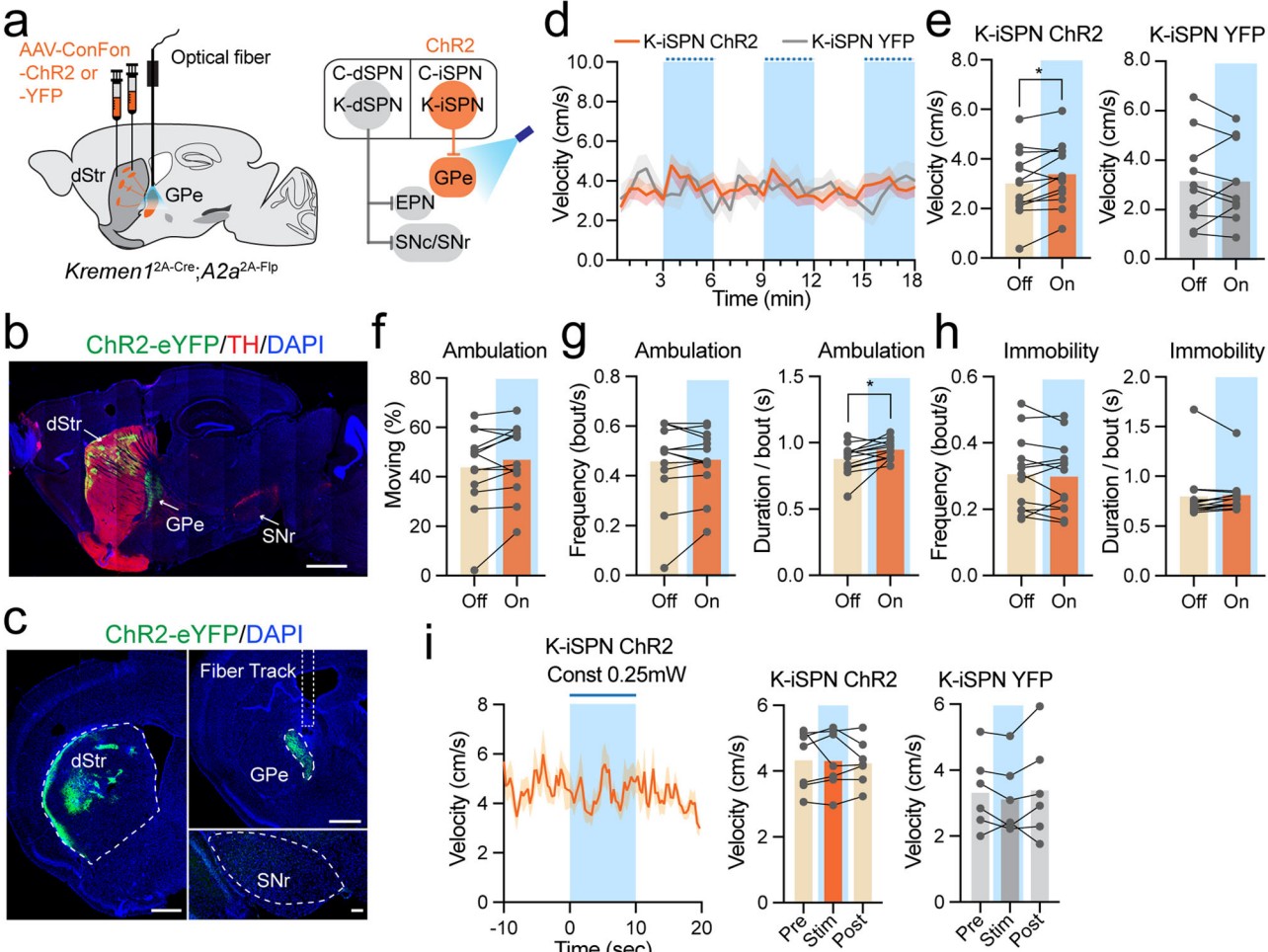

**Fig. 6 | Optogenetics activation of *Kremen1*[+] iSPNs leads to a modest increase of locomotion. a** Schematic showing ChR2 injection and optical fiber implantation for activating *Kremen1*[+] iSPNs axons in the GPe in *Kremen1*[2A-Cre];*A2a*[Flp] mice. **b, c** Representative images of sagittal section (**b**) and coronal sections (**c**), illustrating ChR2 (green) expression in *Kremen1*[+] iSPNs. Scale bars: 1 mm (sagittal section), 500 μm (dStr and GPe) and 100 μm (SN). **d** Instantaneous locomotion velocity during the open-field test comparing light-off and light-on (blue shaded area) conditions. ChR2 = 13 mice, YFP = 10 mice. Mean (solid line) ± SEM (shaded area). **e** Average velocity during light-off (Off) and light-on (On) periods. ChR2 group: $t_{(12)} = 2.85$, paired t-test, two-tailed, *$p = 0.015$, $n = 13$ mice. YFP group: $t_{(9)} = 0.06$, paired t-test, two-tailed, $p = 0.95$, $n = 10$ mice. Data are shown for individual mice and as population averages. **f** The percentage of total ambulation time during light-

off and light-on from ChR2 group: $t_{(12)} = 1.84$, paired t-test, two-tailed, $p = 0.09$. **g** Frequency (left, $t_{(12)} = 0.42$, paired t-test, two-tailed, $p = 0.68$) and duration (right, $t_{(12)} = 2.82$, paired t-test, two-tailed, *$p = 0.016$) of ambulation bout in ChR2 group, $n = 13$ mice. **h** Frequency (left, $t_{(12)} = 0.65$, paired t-test, two-tailed, $p = 0.53$) and duration (right, $t_{(12)} = 0.53$, paired t-test, two-tailed, $p = 0.61$) of immobility bout in ChR2 group, $n = 13$ mice. **i** Instantaneous velocity aligned to 0.25 mW/constant light stimulations (10 s) of *Kremen1*[+] iSPNs (left). Mean (solid line) ± SEM (shaded area). Average velocity during pre-stimulation (Pre), stimulation (Stim), and post-stimulation (Post) for the ChR2 group (middle) and the YFP group (right). For the ChR2 group: one-way ANOVA, $F(1.868, 11.21) = 0.14$, $p = 0.85$, $n = 7$ mice; for the YFP group: one-way ANOVA, $F(1.577, 7.886) = 1.22$, $p = 0.33$, $n = 6$ mice. All error bars were represented as mean ± SEM.

(IACUC) of the Intramural Research Program of the National Institute on Aging (NIA), NIH. All mouse lines were maintained as heterozygotes in a C57BL/6 J background. The *Kremen1*[2A-Cre] KI mice were generated by Shanghai Model Organisms Inc. (Shanghai, China). The *Aldh1a1*[CreERT2] KI mice were generated as previously described[14]. The *Nr4a1*−eGFP (Stock No: 036737-UCD) transgenic mice were obtained from Mutant Mouse Resource & Research Centers (MMRRC). The *Calb1*[IRES2-Cre] mice (Stock No: 028523) and Ai14 (Stock No: 007908) were obtained from the Jackson laboratory. The *A2a*[2A-Flp] KI mice were generated by Dr. Gerfen and the Rodent Transgenic Core of National Institute of Mental Health (NIMH). Both females and males were used for all experiments. Mice used for viral injections were between 2 and 4 months of age. The mice were housed in a 12-hour-light/12-hour-dark cycle in groups of 2-5 animals and had ad libitum water and a regular diet. All the behavioral tasks were performed during the light cycle. Littermates were randomly assigned to different groups prior to experiments.

## Generation of *Kremen1*[2A-Cre] KI mice
The *Kremen1*[2A-Cre] KI mice were generated using the CRIPSR/Cas9 approach in the C57BL/6 J strain by Shanghai Model Organisms Inc. A donor plasmid containing the 2 A ribosome skipping sequence, Cre DNA recombinase sequence, and flanking *Kremen1* mouse genomic DNA sequence was constructed. This was used to insert to the 2A-Cre DNA fragment into exon 9 immediately after the stop codon of *Kremen1* gene locus, guided by the gRNA with sequence GTGGGCTTCAGTCACTCACG AGG. One founder mouse was generated, and the correct genomic modification was confirmed by sequencing.

## Laser capture microdissection and RNA sequencing
One-month-old *Nr4a1-eGFP* transgenic mice were euthanized with $CO_2$ followed by rapid decapitation. The brains were immediately dissected and preserved at −80 °C. Cryosectioning of the frozen brains was performed at −20 °C, and the sections were mounted onto PAN membrane frame slides (Applied Biosystems, Foster City,

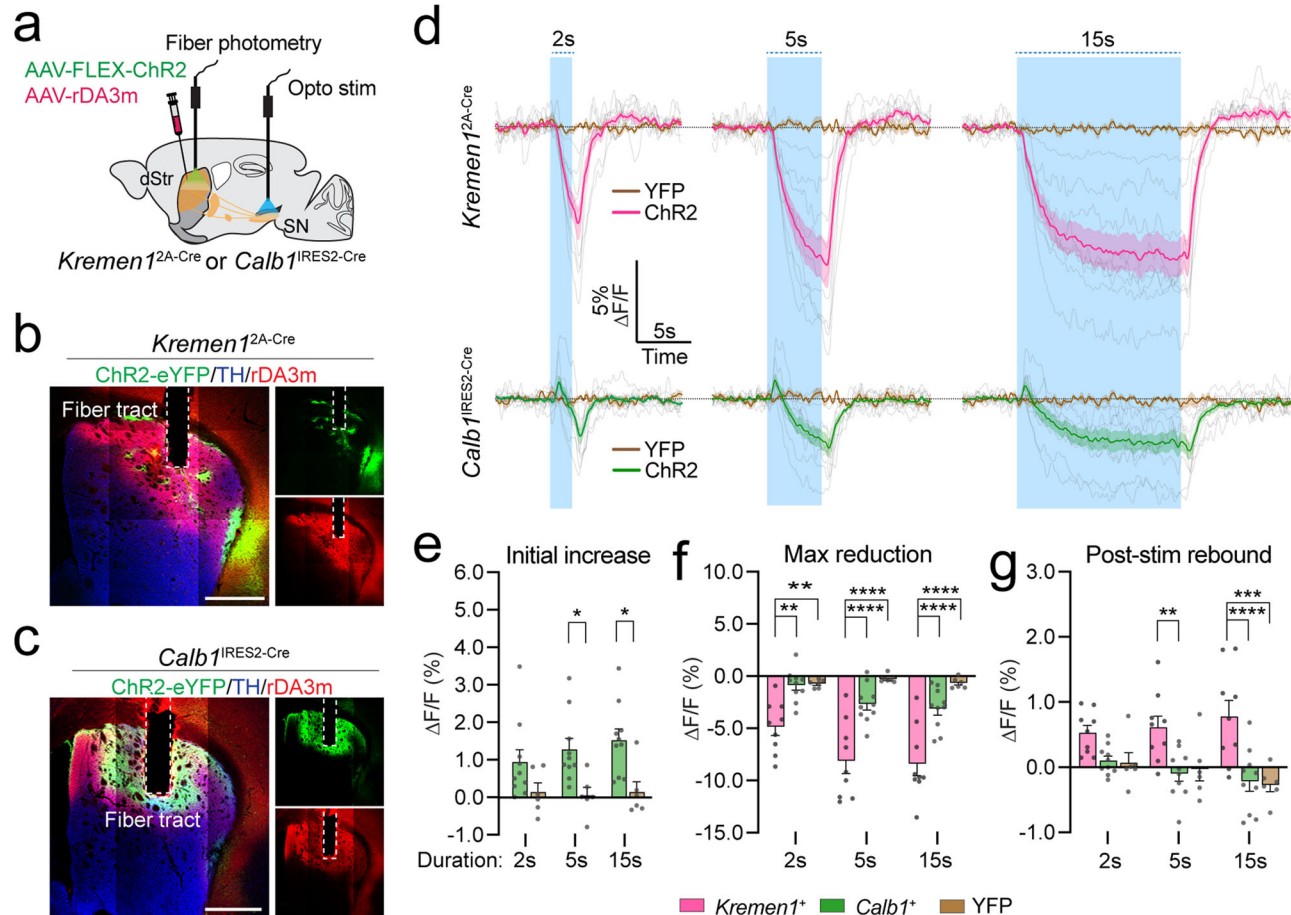

**Fig. 7 | *Kremen1*+ and *Calb1*+ dSPNs differentially regulate dopamine release.**
**a** Schematic of simultaneous fiber photometry and optogenetics to assess dopamine changes in the dorsal striatum in response to *Kremen1*+ and *Calb1*+ dSPN activation. **b**, **c** Representative images showing ChR2 (green), rDA3m (red), and TH (blue) in the dorsal striatum of *Kremen1*$^{2A-Cre}$ (**b**) and *Calb1*$^{IRES2-Cre}$ (**c**) mice. Scale bar: 500 μm. **d** Dopamine signal changes during 20 Hz light stimulation of SN for varying durations in *Kremen1*$^{2A-Cre}$ (top) and *Calb1*$^{IRES2-Cre}$ (bottom) mice. Solid traces with shade represent population mean ± SEM of ΔF/F changes aligned to stimulation onset. Light gray traces show mean ΔF/F changes for individual *Kremen1*- ChR2 ($n = 9$ mice) and *Calb1*- ChR2 ($n = 10$ mice) mice, with YFP group (*Kremen1*-YFP, $n = 3$ mice; *Calb1*-YFP, $n = 3$ mice). Same mouse dataset was used for further analysis in (**e**–**g**). **e** Initial dopamine elevation amplitudes following *Calb1*+ dSPNs activation compared to YFP controls. Two-way ANOVA: activation effect $F(1,42) = 19.44$,

$^{****}p = 7.1e-5$; duration effect $F(2,42) = 0.4331$, $p = 0.6513$; interaction effect $F(2,42) = 0.4730$, $p = 0.6264$. Multiple comparisons: 5 s, $^*p = 0.0258$; 15 s, $^*p = 0.0102$. **f**, **g** Maximum dopamine reduction amplitudes (**f**) and post-stimulation (post-stim) dopamine release changes (**g**) following activation of *Kremen1*+ and *Calb1*+ dSPNs compared to YFP controls. **f** Two-way ANOVA: activation effect $F(2,66) = 54.47$, $^{****}p = 1.1e-14$; duration effect $F(2,66) = 4.723$, $^*p = 0.0121$; interaction effect $F(4,66) = 1.575$, $p = 0.1914$. Multiple comparisons, *Kremen1*-ChR2 vs. *Calb1*-ChR2: 2 s, $^{**}p = 0.0025$; 5 s, $^{****}p = 1.95e-6$; 15 s, $^{****}p = 3.3e-6$; *Kremen1*-ChR2 vs. YFP: 2 s, $^{**}p = 0.0087$; 5 s, $^{****}p = 9.2e-9$; 15 s, $^{****}p = 1.12e-8$. **g** Two-way ANOVA: activation effect $F(2,66) = 21.66$, $^{****}p = 5.9e-8$; duration effect $F(2,66) = 0.5153$, $p = 0.5997$; interaction effect $F(4,66) = 1.234$, $p = 0.3052$. Multiple comparisons, *Kremen1*-ChR2 vs. *Calb1*-ChR2: 5 s, $^{**}p = 0.0095$; 15 s, $^{****}p = 9.06e-6$; *Kremen1*-ChR2 vs. YFP: 15 s, $^{***}p = 0.0003$. All data are represented as mean ± SEM.

CA). The dorsal striatum was determined based on anatomic landmarks such as the corpus callosum, lateral ventricle and nucleus accumbens. Using the ArturusXT microdissection system with fluorescent illumination (Applied Biosystems), the eGFP-positive island-like structures within the dorsal striatum of *Nr4a1*-eGFP transgenic mice were carefully isolated onto LCM Macro Caps (Applied Biosystems) and designated as "patch", while surrounding tissues of similar size were also isolated onto Macro Caps and designated as "matrix".

Total RNA extraction and purification were executed from hundreds of caps using the PicoPure Isolation kit (Applied Biosystems), with subsequent genomic DNA clearance facilitated by RNase-free DNase (Qiagen) following the manufacturer's protocols. Quantification of RNA was performed using a NanoDrop spectrophotometer (ThermoFisher), and RNA integrity was evaluated using the Bioanalyzer RNA 6000 pico assay (Agilent). Only RNA samples exhibiting high integrity, as indicated by ribosomal integrity numbers (RIN) of 7 or higher, were selected for subsequent patch and matrix RNAseq library preparation.

The cDNA libraries were prepared from the purified RNA using the TruSeq Stranded Total RNA LT library preparation kit (Illumina) following the manufacturer's protocol. The quality of the libraries was assessed using the Bioanalyzer DNA 1000 assay (Agilent) before sequencing on an Illumina HiSeq 2000 platform. Fastq files were generated using the standard Illumina pipeline. Transcript abundance, annotated by Ensembl, was quantified using Salmon in a non-alignment-based mode, and gene-level counts were estimated utilizing the Tximport package (Bioconductor). Normalization of counts and subsequent data analysis were conducted following previously established procedures[26]. The accession number of the striatal tissue RNA-seq is PRJNA870469.

### RNA in situ hybridization and image analysis
RNA in situ hybridization (ACDBio, RNAscope) was used to detect the expression of *Drd1*, *Drd2*, *Kremen1* and *Tshz1* mRNAs in the dorsal striatum of adult C57BL/6 J mice. For tissue preparation, mice were anesthetized with $CO_2$ and rapidly decapitated. The brains were fresh-

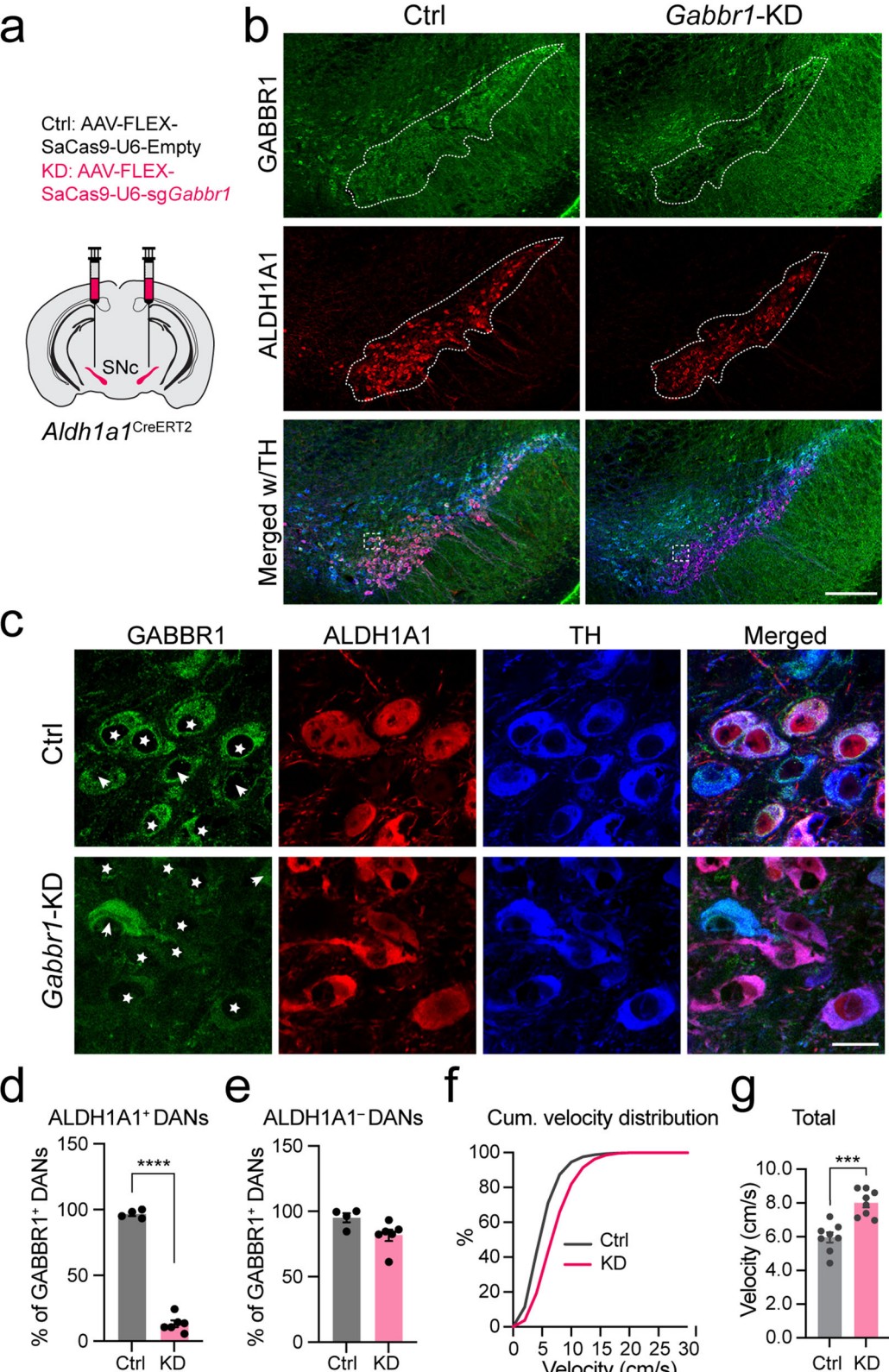

**Fig. 8 | Knockdown of *Gabbr1* receptor in *Aldh1a1*+ DANs leads to hyperactivity.**
**a** Schematic illustrating *Gabbr1*-KD or control (Ctrl) AAV injection in the SNc of *Aldh1a1*CreERT2 mice. **b** Representative images showing GABBR1 (green), ALDH1A1 (red) and TH (blue) staining in the SNc of *Gabbr1* Ctrl and KD mice. Dash lines outline areas enriched with *Aldh1a1*+ DANs. Scale bar: 200 μm. **c** High magnification view of boxed regions in (**b**). Stars indicate *Aldh1a1*+ DANs, and arrows highlight *Aldh1a1*– DANs. Scale bar: 20 μm. **d** Percentage of GABBR1+ neurons in *Aldh1a1*+ DANs in the SNc. *Gabbr1* Ctrl: 96.71 ± 1.49%; *Gabbr1* KD: 13.22 ± 2.58 %. Unpaired t-test, two-tailed, ****$p$ = 8.6e-9; $n$ = 6 mice (KD), $n$ = 4 mice (Ctrl). **e** Percentage of GABBR1+ neurons in *Aldh1a1*– DANs in the SNc. *Gabbr1* Ctrl: 95.10 ± 3.60 %; *Gabbr1* KD: 81.76 ± 4.38 %. Unpaired t-test, two-tailed, $p$ = 0.063; $n$ = 6 mice (KD), $n$ = 4 mice (Ctrl). **f** Cumulative frequency distribution of velocity for *Gabbr1* Ctrl and KD mice, $n$ = 8 mice per group. **g** Average velocity during the 30-min test period. Unpaired t-test, two-tailed, ***$p$ = 0.0002, $n$ = 8 mice per group. All data are presented as mean ± SEM.

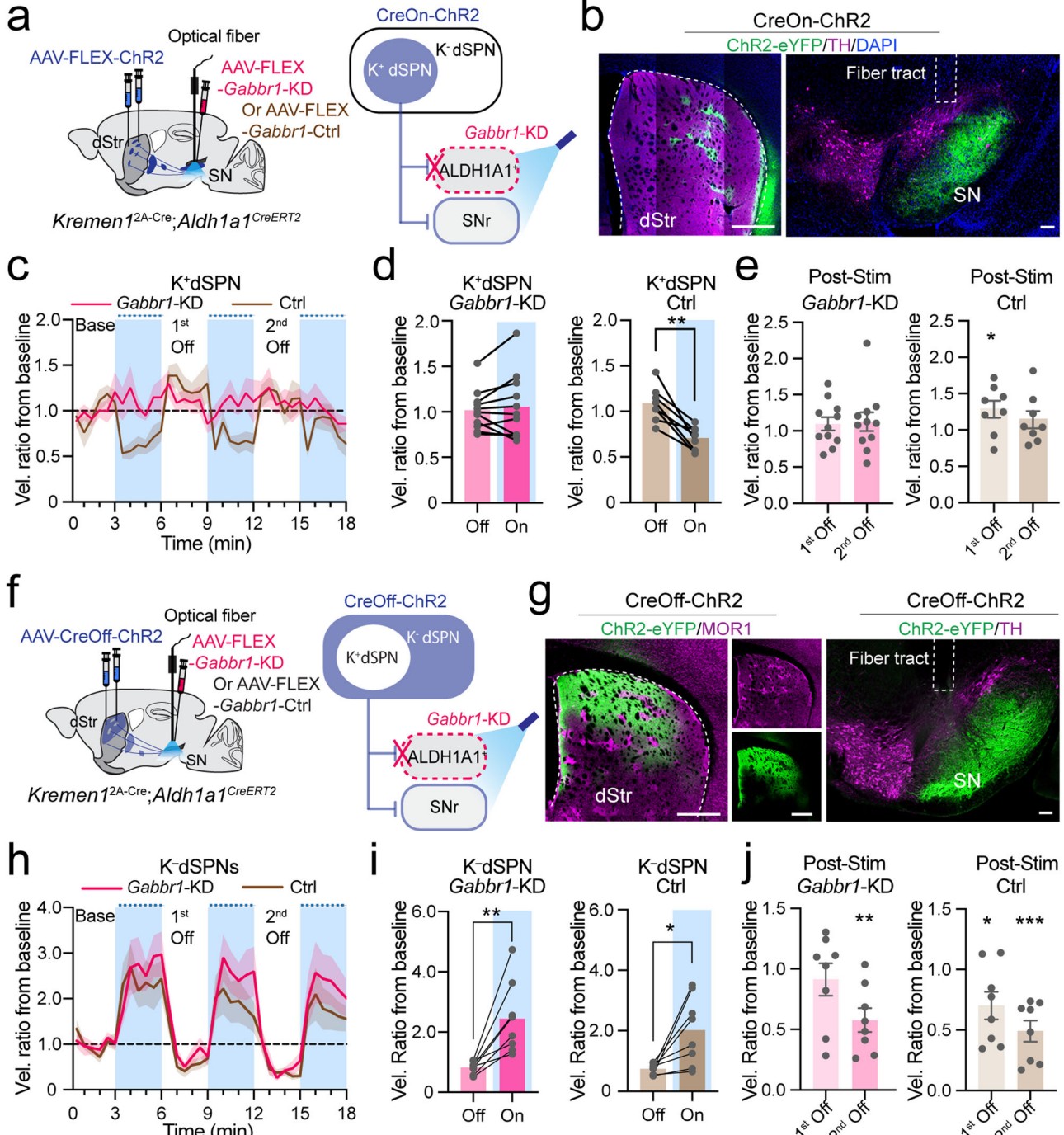

**Fig. 9 | Knockdown of *Gabbr1* receptors in *Aldh1a1*⁺ DANs abolishes *Kremen1*⁺ dSPN-induced locomotion alterations. a** Schematic showing selective activation of *Kremen1*⁺ dSPNs (K⁺dSPN, **a**) or *Kremen1*-negative dSPNs (K⁻dSPN, **f**) and genetic deletion of *Gabbr1* receptors (*Gabbr1*-KD) in *Aldh1a1*⁺ DANs in *Kremen1*²ᴬ⁻ Cre;*Aldh1a1*ᶜʳᵉᴱᴿᵀ² double KI mice. **b** Representative images of ChR2 (green), TH (magenta) and DAPI (blue) in the dorsal striatum and SN. Scale bars: 500 μm (dStr) and 100 μm (SN). **c** Changes in instantaneous locomotion velocity (relative to the first 3-min baseline) during open-field test with K⁺dSPN activation under *Gabbr1*-KD and control (Ctrl) conditions. *Gabbr1*-KD: *n* = 11 mice; Ctrl: *n* = 8 mice, the same mouse dataset was used in (**c**–**e**). **d** Comparison of velocity changes between light-off and -on periods. *Gabbr1*-KD: t₍₁₀₎ = 0.72, paired t-test, two-tailed, *p* = 0.49. Ctrl (*n* = 8 mice): t₍₇₎ = 5.40, **p* = 0.001. **e** Velocity changes during the 1ˢᵗ Off and 2ⁿᵈ Off periods in *Gabbr1*-KD and Ctrl. One sample t-test, two-tailed. *Gabbr1*-KD: 1ˢᵗ Off, *p* = 0.32; 2ⁿᵈ

Off, *p* = 0.35. Ctrl: 1ˢᵗ Off, *p* = 0.045, 2ⁿᵈ Off *p* = 0.27. **g** Representative images of ChR2 (green), MOR1 (magenta) in the dorsal stratum, and ChR2 (green), TH (magenta) in the SN. Scale bars: 500 μm (dStr) and 100 μm (SN). **h** Changes in instantaneous locomotion velocity (relative to the first 3-min baseline) during open-field test with K⁻dSPN activation under *Gabbr1*-KD and Ctrl conditions. *Gabbr1*-KD: *n* = 8 mice, Ctrl: *n* = 8 mice, the same mouse dataset was used in (**h**–**j**). **i** Comparison of velocity changes between light-off and -on periods. *Gabbr1*-KD: t₍₇₎ = 4.11, paired t-test, two-tailed, **p* = 0.0045. Ctrl: t₍₇₎ = 3.26, **p* = 0.014. **j** Velocity changes during the 1ˢᵗ Off and 2ⁿᵈ Off periods in *Gabbr1*-KD and Ctrl mice. One sample t-test, two-tailed. *Gabbr1*-KD: 1ˢᵗ Off, *p* = 0.54; 2ⁿᵈ Off, **p* = 0.0035. Ctrl: 1ˢᵗ Off, **p* = 0.035; 2ⁿᵈ Off, ***p* = 0.0006. All error bars are presented as mean ± SEM. The data are presented as mean (solid line) ± SEM (shaded area) in velocity traces.

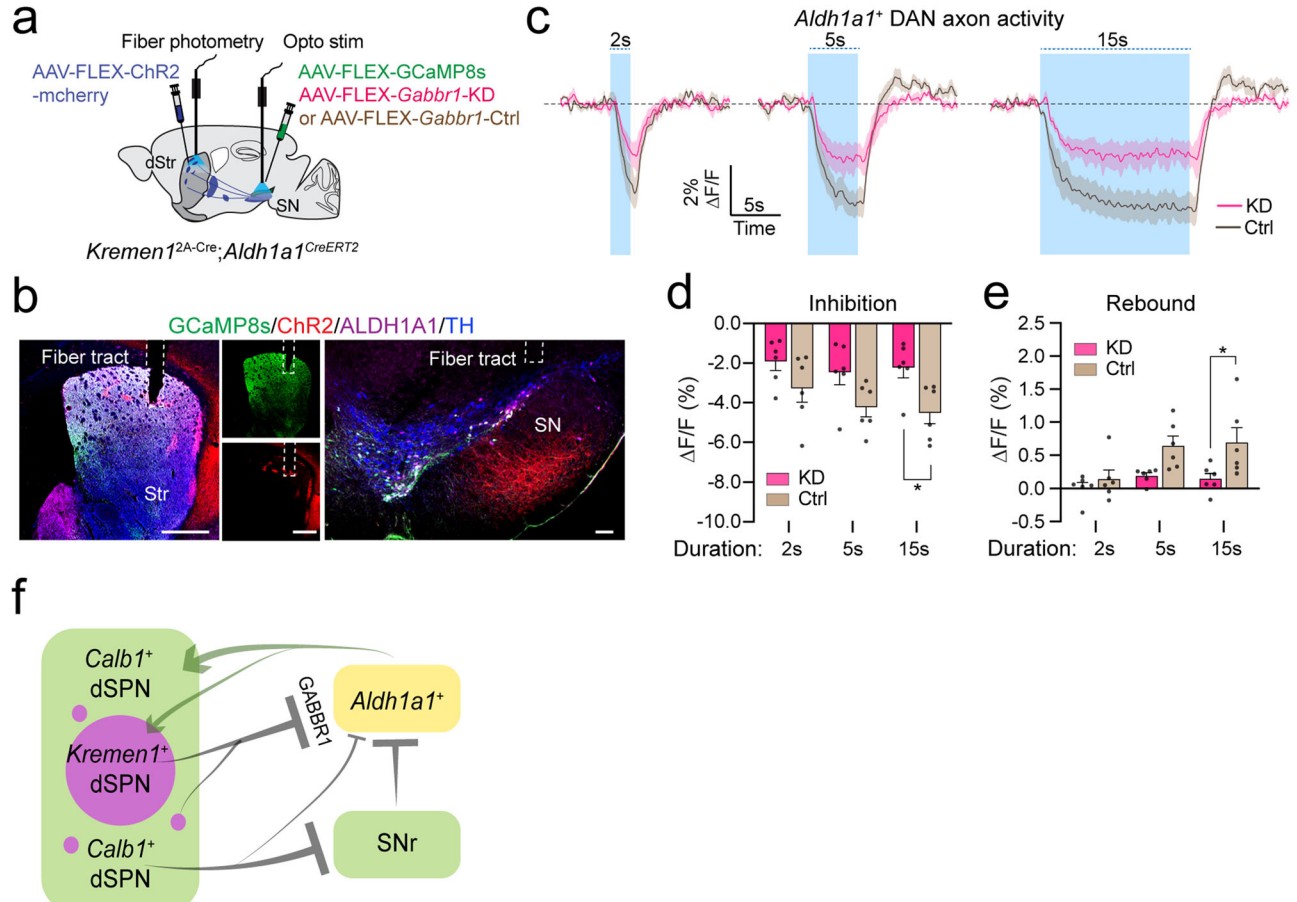

**Fig. 10 | Genetic knockdown of *Gabbr1* receptor affects *Kremen1*⁺ dSPN-induced alterations in *Aldh1a1*⁺ DAN activity. a** Schematic illustrating simultaneous fiber photometry recording of *Aldh1a1*⁺ DAN activity and optical activation of *Kremen1*⁺ dSPNs in *Aldh1a1*⁺ DAN-specific *Gabbr1* control (Ctrl) and knockdown (KD) mice. **b** Representative images showing ChR2 (red), GCaMP8s (green), ALDH1A1 (magenta) and TH (blue) staining in the Str of *Kremen1*²ᴬ⁻ᶜʳᵉ;*Aldh1a1*ᶜʳᵉᴱᴿᵀ² double KI mice. Optical fiber locations in the striatum and SN are indicated. Scale bars: 500 μm (Str) and 100 μm (SN). **c** Changes in GCaMP8s signals in the *Aldh1a1*⁺ DAN axon terminals in the dStr during 20 Hz light stimulation at varying durations of *Kremen1*⁺ dSPN activation in the SN under *Gabbr1*-KD (red) and Ctrl (brown) conditions. Data are presented as mean of each group of mice (solid line) ± SEM (shaded area). *n* = 6 mice per each group. **d** Amplitude of *Aldh1a1*⁺ DAN activity

reduction following activation of *Kremen1*⁺ dSPNs in KD and Ctrl conditions. *n* = 6 mice per group. Two-way ANOVA: stimulation duration effect, $F_{(2, 30)} = 1.202$, *p* = 0.3147; genotype effect, $F_{(1, 30)} = 15$, ***p* = 0.0005; interaction effect, $F_{(2, 30)} = 0.3312$, *p* = 0.72. Multiple comparisons test: *p* = 0.024. **e** Post-stimulation rebound of *Aldh1a1*⁺ DAN activity following activation of *Kremen1*⁺ dSPNs in KD and Ctrl conditions. *n* = 6 mice per group. Two-way ANOVA: stimulation duration effect, $F_{(2, 30)} = 4.4651$, *p* = 0.02; genotype effect: $F_{(1, 30)} = 12.20$, ***p* = 0.0015; interaction effect, $F_{(2, 30)} = 1.408$, *p* = 0.26. Multiple comparisons test: *p* = 0.0185. All error bars are represented as mean ± SEM. **f** The schematic summary illustrates that *Calb1*⁺ SPNs promote locomotion, whereas *Kremen1*⁺ SPNs terminate ongoing movement by inhibiting the activity of *Aldh1a1*⁺ DANs via GABBR1 receptors.

frozen on dry ice and stored at -80 °C before sectioning. Striatal sections (12 μm) were collected using a cryostat (Leica Biosystems) and stored at -80 °C until processed.

RNAscope was conducted according to the instructions of RNAscope Multiplex Fluorescent Reagent Kit v2 user manual. Probes against *Drd1* (Cat. No. #401901), *Drd2* (Cat. No. #406501), *Kremen1* (Cat. No. #425771) and *Tshz1* (Cat. No. #494291) were applied to brain sections. Fluorescent images were acquired using a laser scanning confocal microscope LSM 780 (Zeiss) with 20× or 40× lens. Bregma coordinates around 1.34 mm, 0.92 mm, 0.5 mm, 0.08 mm and -0.34 mm were selected for image analysis.

RNAscope images were analyzed using Imaris (v10.0.0, Bitplane, Belfast Northern Ireland, UK). Surfaces for the dorsal and ventral striatum were created using the Allen Brain Atlas as a reference. The spatial patch was defined by outlining *Kremen1*⁺ SPN clusters (a minimum of 5 SPNs), and the density of *Kremen1*⁺ SPNs within a spatial patch is at least 200 cells/mm². Surfaces for individual channels (i.e., *Drd1*, *Drd2*, *Kremen1*, *Tshz1* and DAPI) were created with unique

parameters, which were saved and applied to all images within the same batch. Minor adjustments were made in subsequent rounds of RNAscope to improve quantification accuracy. DAPI surfaces in the striatum were filtered by the mean intensity of *Drd1* (dSPNs) and the mean intensity of *Drd2* (iSPNs). These *Drd1*⁺ and *Drd2*⁺ cells were further filtered by the overlapped volume ratio to *Kremen1* to quantify patch dSPNs and patch iSPNs, respectively. RNAscope analysis with *Tshz1* was conducted in a similar manner, filtering *Drd1*⁺ and *Kremen1*⁺ cells by the overlapped volume ratio to the *Tshz1* surface. All data points are presented as the average across striatal sections of both hemisphere for each mouse.

Due to the limitation of staining a maximum of three probes simultaneously using RNAscope V2 reagents, we stained different batches of brains with *Kremen1*, *Drd1*, and *Drd2* probes for Fig. 1d, and with *Kremen1*, *Tshz1*, and *Drd1* probes for Supplementary Fig. S1. The slightly difference in the numbers arise because the data were generated from two separate batches of brains and counted by different individuals.

## Immunohistochemistry

Mice were anesthetized with pentobarbital and transcardially perfused with precooled PBS, followed by 4% paraformaldehyde (PFA) solution as described previously[14,35]. Brains were stored in 4% PFA at 4 °C overnight and then transferred to 30% PBS buffered sucrose solution for at least 2 days before sectioning. Series of coronal sections (40 μm) were collected using a cryostat and stored at 4 °C in PBS. Sections were blocked in 10% normal donkey serum (Sigma-Aldrich) + 0.5% Bovine serum albumin (Sigma-Aldrich) + 0.3% Triton for 1 hr at room temperature (RT). Next, sections were incubated with the primary antibodies over one or two nights at 4 °C. Sections were then washed with PBS (3 × 10 min) and incubated with fluorescent secondary antibodies at RT for 1 hr. In some experiments, sections were further washed twice in PBS, incubated with DAPI (40, 6-diamidino-2-phenylindole, Invitrogen, D1306) (0.5 mg/ml in PBS) for 1 min. After washing with PBS (5 min), sections were mounted onto subbed slides, and coverslipped with mounting media (ProLong® Gold Antifade Mountant, Life technology). Images were taken using a laser scanning confocal microscope LSM 780 (Zeiss). The primary antibodies used for immunostaining included rabbit monoclonal anti-TH (Pel-Freez Biologicals, P40101; dilution 1:1000), mouse monoclonal anti-TH (ImmunoStar, 22941; dilution 1:1000), chicken polyclonal anti-TH (Aves Labs, TYH; dilution 1:500), rabbit anti-Mu-opioid receptor (MOR) (ImmunoStar, 24216; dilution 1:3000), chicken polyclonal anti-GFP (Aves Labs, GFP-1020; dilution 1:1000), mouse polyclonal anti-RFP (Rockland, 200-301-379; dilution 1:1000), guinea pig anti-Parvalbumin (Swant, GP72; dilution 1:1000), rabbit anti-Somatostatin-14 (T-4103, Peninsula Laboratories; dilution 1:1000), goat anti-ChAT (AB144P, Millipore; dilution 1:500), rabbit anti-Iba1 (Wako, 019-19741, 1:1000), rabbit anti-GFAP (Abcam, ab7260; dilution 1:2000), goat anti-CD13 (R&D Systems, AF2335; dilution 1:100), and mouse monoclonal anti-GABA$_B$ (Abcam, ab55051, dilution 1:500). Appropriate fluorophore-conjugated secondary antibodies (Life Technologies) were used depending on the desired fluorescence colors.

## Neuron counting in *Gabbr1* KD mice

Coronal midbrain sections from *Gabbr1* KD and Ctrl mice (40 μm thickness, approximately four sections per mouse spanning Bregma −2.70 mm to −3.80 mm) were stained with anti-TH, anti-ALDH1A1 and anti-GABBR1 antibodies for quantification. Images of the SN were captured as single layers by a laser scanning confocal microscope (LSM 780, Zesis) with × 20 objective lens. The percentage of GABBR1 receptor-positive neurons in ALDH1A1$^+$/TH$^+$ and ALDH1A1$^-$/TH$^+$ dopaminergic neurons was qualified using ImageJ (v1.53t). The SNc region was manually delineated based on previously described method[14].

## Viral vectors

The following AAV vectors were purchased from Addgene (Watertown, MA, USA): AAV9-hSyn-DIO-hM4D(Gi)-mCherry (#44362), AAV9-hSyn-DIO-mCherry (#50459), AAV1-EF1a-double floxed-hChR2(H134R)-EYFP (#20298), AAV1-hSyn-hChR2(H134R)-EYFP (#26973), AAV8-hSyn-Con/Fon-hChR2(H134R)-EYFP (#55645), AAV8-hSyn-Con/Fon-EYFP (#55650), AAV1-Ef1a-DIO-EYFP (#27056), and AAV1-syn-FLEX-jGCaMP8s (#162377). AAV2/9-hSyn-DIO-GRAB-rDA3m were produced by BrainVTA (Gaithersburg, MD, USA). AAV9-FLEX-SaCas9-U6-sgGabbr1 was packaged by SignaGen Laboratories (Frederick, MD, USA). AAV9-CMV-NLS-SaCas9-3XHA-sNRPpa-U6-Basal-empty was packaged by Vigene Biosciences (Rockville, MD, USA).

## Construction of pAAV-FLEX-SaCas9-U6-sg*Gabbr1*

The reference sequence of *Gabbr1* gene was retrieved from the UCSC genome browser database (http://genome.ucsc.edu/), and subsequent identification and alignment of exons were performed using the Mouse Genome Informatics (MGI) database. The most 5′ common coding exons were then selected, and the sequence was uploaded to the CRISPOR website (http://crispor.org) for identifying potential sgRNAs and PAM sequences. Three *Gabbr1* sgRNAs were synthesized as short oligos (Eurofins Genomics) with a 5′ CACC- 3′ overhang on the forward primer, and a 5′ -AAAC 3′ overhang on the reverse primer, facilitating seamless integration into the pX601-AAV-CMV;NLS-SaCas9-NLS-3xHA-bGHpA;U6;BsaI-sgRNA vector, a gift provided by Dr. Feng Zhang (Addgene plasmid # 61591). Sanger sequencing confirmed the insertion of the *Gabbr1* sgRNA using the primer TAACCACGTGAGGGCCTATTTC.

Mouse Neuro-2a (N2a) cell lines were cultured in Dulbecco's modified Eagle's medium (DMEM) supplemented with 10% FBS (HyClone) at 37 °C with 5% CO$_2$ incubation. Transfection of Cells with the plasmid pX601-AAV-CMV;NLS-SaCas9-NLS-3xHA-bGHpA;U6;sg*Gabbr1* vector was performed using Neuro-2a Transfection Kit (Altogen Biosystems), following the manufacturer's protocol. Cells were then harvested for PCR-based identification of mutations induced by genome editing using the Guide-it Mutation Detection Kit (Cat. No. 631443). The sg*Gabbr1* with the sequence of CACCGCGCCACACTCCACAATCCCAC exhibited high efficiency in genome editing under SaCas9 and was thus selected for further integration into the pAAV-Flex-SaCas9-U6-sg*Daglb* vecgtor[26]. The primers 5′-TAACCACGTGAGGGCCTATTTCCCATGATT and 5′-CG CTCGGTCCGAAAAATCTCGCCAACAAGTT were used to amplify the fragment containing sg*Gabbr1*, which replaced sg*Dalgb* in the plasmid pAAV-Flex-SaCas9-U6-sg*Daglb* at the *PmlI* and *RsrII* registration sites. Sanger sequencing confirmed the presence of sg*Gabbr1* in the resulting pAAV-Flex-SaCas9-U6-sg*Gabbr1* plasmid.

## Stereotaxic injection and optic fiber implantation

The stereotaxic survival surgery was performed as previously described[26]. All surgery were conducted under aseptic conditions, and body temperature was maintained using a heating pad. Briefly, adult mice (2-4 months old) were anesthetized with isoflurane (1-2%) and head-fixed in a stereotaxic frame (Kopf Instruments). A total volume of 500-700 nL of AAVs was injected unilaterally or bilaterally into the dorsal striatum with two locations per hemisphere (coordinates: 0.5 mm AP to Bregma, ±2.2 mm ML, −2.5 mm DV from dura surface; 1.3 mm AP from Bregma, ±1.8 mm ML, −2.5 mm DV from dura surface) or SN (coordinates: −3.1 mm AP from Bregma, ±1.5 mm ML, −3.9 mm DV from dura surface). The infusion of viruses was controlled by a stereotaxic injector (Stoelting) at a speed of 75 nL/min. After a 5-minute wait following the end of injection, the injector was slowly withdrawn. The scalp was then sutured, and the mice were returned to their home cages. All behavior experiments were performed at least 4 weeks after injection to allow time for full heterologous gene expression. For *Aldh1a1*$^{CreERT2}$ mice, 4-OHT was injected intraperitoneally at a dosage of 10 mg/kg bodyweight for five consecutive days, beginning one week after surgery, to induce *Cre* recombinase expression.

To prepare mice for optogenetics or fiber photometry experiments, we performed a second surgery to implant optic fibers at least 3 weeks after viral injection. For behavioral optogenetics experiments, optical fiber stubs (200 μm core, 0.39 NA, Thorlabs) were bilaterally implanted with the tips positioned over the SNc (−3.1 mm AP from Bregma, ±1.5 mm ML, −3.9 mm to −4.1 mm DV from dura surface) or GPe (−0.3 mm AP from Bregma, ±2.0 mm ML, −3.5 mm from dura surface). The optic fibers were secured in place with a thick layer of radiopaque adhesive cement (C&B METABOND, Parkell). Once dried, Vetbond tissue adhesive (3 M) was applied to seal the head incision with adhesive cement.

For simultaneous photometry recordings and optogenetics experiments, optic fiber stubs with a 200 μm core and 0.39 NA (Thorlabs) were unilaterally implanted over the SN (− 3.1 mm AP from Bregma, 1.5 mm ML, −3.9 mm to −4.1 mm DV from dura surface) for optogenetic stimulation. In addition, optic fiber stubs with a 200 μm core and 0.5 NA (Plexon) were implanted over dorsolateral striatum

(+1.0 mm AP from Bregma, 2.0 mm ML, −2.2 mm to −2.5 mm DV from the dura surface) on the same side for striatal rDA3m photometry recording during optogenetic stimulation.

For the treadmill fiber photometry experiment, optic fiber stubs with a 200 μm core and 0.5 NA (Thorlabs) were unilaterally or bilaterally implanted in the SN (−3.1 mm AP from Bregma, 1.5 mm ML, −4.0 mm to −4.3 mm DV from dura surface) to record the axon calcium signals of patch or matrix dSPNs. The optic fibers were fixed in place using adhesive cement. A titanium metal head-bar (Labeotech) was mounted on top of the adhesive cement rostral to the fiber stubs for head-restraint. The animals were allowed to recover for at least one week after the optic implantation before any optogenetics or fiber photometry experiments.

## Open-field spontaneous locomotion
Locomotion in freely moving mice was measured using video tracing analyses. Mice were habituated for 30 min in the behavioral room, wherein a 20 W lamp was shielded in a box and positioned in the dark opposite to the testing area to provide a diffuse light source. For the video tracing test, mice were placed in a 50 cm × 50 cm gray opaque chamber for 30 min. Activity was recorded from overhead by a digital camera at a frame rate of 15 Hz. EthoVision XT software (Noldus) was used to track the mice and analyze the video for velocity, time, and distance traveled.

## Chemogenetic manipulation
JHU37160-dihydrochloride (Hello Bio) was dissolved in water to achieve a stock concentration of 0.3 mg/mL and stored in small aliquots at -20 °C. Prior to each use, the working solution was freshly prepared by diluting the stock 10-fold with 0.9% saline. Mice received 0.3 mg/kg bodyweight dosage via intraperitoneal injection 30 minutes before behavioral tests.

## Optogenetic stimulation
A LED light source with commutator (PlexBright, Plexon) was connected to a multimodal optic fiber patch cable (200 μm core, 0.39 NA, Plexon) through ceramic sleeves (Thorlabs) to the ferrule of the optic fiber stubs previously implanted in the mouse. Light power was calibrated using a power meter (PM100D, Thorlabs) to achieve the desired output measured at the tip of optic fiber. For ChR2 experiments, photo stimulation consisted of 5ms-width blue light pluses (465 nm, 3 mW) at different frequency or constant blue light (465 nm, 0.25 mW). Light pulses were generated by a TTL pulse generator (OPTG-4, Doric Lenses).

## Optogenetics with open field
Mice were habituated in the testing room 30 min before testing, and the apparatus was cleaned with 50% ethanol between animals. Mice were placed in a 50 cm × 50 cm clear chamber and their activities were captured by top-view and side-view cameras (Logitech). After a 3 min exploration period, mice received either 3 min ON and 3 min OFF bilateral stimulation or 10 s ON and 1 min OFF bilateral stimulation. Video and LED light TTL were acquired and synchronized with Synapse software (Tucker-Davis Technologies). Distanced traveled and velocity during the acquisition period were calculated using EthoVision XT software. In Figs. 4 and 9, baseline or prior was identified as the 3 min period prior to 1st stimulation, 1st Off indicates 3 min period post 1st stimulation, and 2nd Off indicates 3 min period post 2nd stimulation. Ambulation bouts were scored as periods of movement >2 cm/s lasting for > 0.5 s and separated by >0.5 s. Immobility bouts were scored as periods of <2% pixel change lasting for >0.5 s and separated by >0.5 s. For behavioral-state dependent effects of optogenetic stimulation experiment, mice received 10 s ON and 1 min OFF bilateral 20 Hz stimulation. Ambulation state was determined by movement velocity >2 cm/s in the entire 0.5 second immediately preceded either stimulation onset or stimulation offset. Quiescence (includes small

movement) state was determined by movement velocity <1.5 cm/s in the entire 0.5 second immediately preceded either stimulation onset or stimulation offset. Due to kinematics of stimulation on behavior, quantification for effects of stimulation on velocity change used 1 s interval instead.

## Optogenetics with fiber photometry
Mice were allowed to freely explore an open field chamber while receiving optical stimulation lasting 2 s, 5 s, or 15 s across approximately 10 trials per stimulation duration, with intervals of at least 45 s between trials. To prevent optic fiber patch cables from tangling, a pigtailed fiber-optic rotary joint (Doric Lens) was used to connect two multimodal optic fiber patch cables in the photometry light path. Realtime rDA3m or GCaMP8s signals and timing of TTL pulses were recorded and synchronized using Synapse software (Tucker-Davis Technologies). For data analysis, ΔF/F changes were normalized by subtracting each sample's own baseline, defined as the average ΔF/F value during the 5 s preceding stimulation onset. The maximum reduction in dopamine or GCaMP amplitude was identified as the lowest ΔF/F value within 1 s interval prior to stimulus offset, representing the peak reduction in response to stimulation. Post-stimulation changes in rDA3m or GCaMP8s signals were calculated as the average ΔF/F during the 5 s interval spanning 4-9 s after stimulus offset, capturing the rebound phase. For $Calb1^{IRES2-Cre}$ mice, the initial dopamine elevation was quantified as the average ΔF/F during the 0.5 s interval between 0.5 s and 1 s after stimulus onset.

## Fiber photometry
Dopamine sensor or GCaMP8s fluorescence was measured using a locked-in amplifier system (Tucker-Davis Technologies, Model RZ10X with Synapse software). The photometry recordings were conducted with either green (GCaMP8s) or red (rDA3m) fluorescence. For GCaMP8s recordings, the blue LED (465 nm) was sinusoidally modulated at 330 Hz, and the UV LED (405 nm) was modulated at 211 Hz as an isosbestic control channel. For rDA3m recordings, the green LED (560 nm) was modulated at 410 Hz, and the UV LED was modulated at 211 Hz. The peak intensity of each LED was calibrated to 20–60 μW, measured at the distal end of the patch cable. Light emissions were filtered through a 6-port fluorescence mixing cube (Doric Lens) before being coupled to an optic patch cable (200μm core, 0.5 NA), affixed to the implanted optic fiber in each mouse. The emitted fluorescent signals were collected by integrated photosensors in RZ10X real-time signal processor equipped with lock-in amplifier. Transistor-transistor logic (TTL) signals were employed to timestamp onset times for each event of interest (e.g., stimulation onset, locomotion onset and offset), which were detected via the RZ10X in the Synapse software. Fiber photometry data was analyzed using custom MATLAB code. Demodulated 465 nm, 560 nm, and 405 nm recording traces were recorded at a sampling rate of 1k Hz.

For the analysis of rDA3m signals during optogenetic stimulation, to capture the large offset change caused by optogenetic stimulation, the demodulated photometry trace of rDA3m was normalized to compute ΔF/F. F was estimated as a referenced rDA3m signal, where a RANSAC ordinary least square linear regression was performed between the demodulated UV isosbestic reference signal and demodulated rDA3m signal to transform the reference signal to account for the differences in gain and offset between the two signals, as well as possible motion artefacts and long-term photo bleaching effects. Then the referenced rDA3m signal (F) was subtracted from the demodulated rDA3m signal to compute ΔF. The same reference approach was also applied to the analysis of GCaMP8s signals of $Aldh1a1^+$ DANs during optogenetic stimulation.

To analyze the modulation of GCaMP8s activity change with locomotor behavior, photometry signal normalization followed a published study[36], using custom MATLAB code. Demodulated

photometry traces of both GCaMP8s and UV isosbestic reference channels were pre-normalized by first computing ΔF/F0. F0 was estimated by calculating the 10th percentile of the raw photometry amplitude using a 5-sec sliding window to account for slow, correlated fluorescence changes, including photobleaching in both channels. ΔF was calculated as raw photometry amplitude subtracted its respective F0. Both channels were initially normalized with this procedure. An additional referencing procedure was performed to remove the effects of motion or mechanical artefacts from analysis. For this referencing procedure, the ΔF/F0 of the UV reference signal was low-pass filtered with a second-order Butterworth filter with a 3 Hz corner frequency. Then a RANSAC ordinary least squares regression between the filtered reference signal and normalized GCaMP8s signal was used to transform the reference signal to account for the differences in gain and offset between the two signals. Lastly, the transformed references trace was subtracted from the normalized GCaMP8s trace as the final ΔF/F. The transformed reference trace was then used as UV ΔF/F. Only the experiments where the maximum percentage of DF/F exceeded 1.5 and the GCaMP8s and the UV reference correlation was below 0.6 were included for further analysis. Z-score of ΔF/F was calculated with mean and standard deviation of the entire recording session.

### Head-fixed voluntary treadmill with fiber photometry

The head-fixed locomotion was achieved with a compact low friction manually driven treadmill, originally designed by Janelia Research Campus of the HHMI, purchased from LABmaker (Berlin, Germany). Teflon belt movement was tracked by a rotary encoder, which sends 0–3.3 V analog output of speed and direction to an ADC port at TDT RZ10x to synchronize with photometry recordings. The speed was calibrated with 10 cm/s at 2.5 V and sampled at 1k Hz. The head fixture was achieved through head fixation system along with an implantable titanium head bar from Labeotech (Montreal, Canada). Mice started treadmill habituation training one week after recovering from head-bar implant surgery to get used to walking on treadmill while head was fixed. Each habituation session lasted 20 min and repeated at most twice a day for 3-4 days until mice were able to spontaneously initiate at least 20 walk bouts during a 20 min session. After habituation training, a 30 min fiber photometry recording session was performed for each individual mouse, while they walked on the treadmill with head fixation. No reward was provided neither during training nor during recording.

Movement versus rest time bins were defined with a 0.25 cm/s threshold on the velocity trace. Isolated movement periods with duration shorter than 0.5 s or movement periods with average velocity smaller than 0.5 cm/s were excluded for analysis. Because mice tended to walk slower on the treadmill, time bins were considered as rest periods only if they lasted longer than 0.8 s. A movement bout initiation time was defined as velocity cross the threshold at the end of the rest period, and a movement termination time was defined as the time velocity fell below the threshold followed by a rest period. Timestamp of maximum value of z-scored GCaMP activity during 1 s before and 1 s after locomotion onset or offset was calculate the time of maximum activity for the respective events. Activity slope during locomotion offset was computed as the coefficient of a least-square linear regression fit for the z-scored GCaMP activity during the interval of 0.5 s to 0 s before locomotion offset. and after locomotion offset. Timestamp of activity rise at locomotion onset was calculated with a 20-ms sliding window to compare with baseline activity which was defined as mean activity during an interval of -1s to -0.5 s preceding the locomotion onset. The 20-ms sliding window started at -0.5 s with a step of 2 ms towards locomotion onset. Mean activity of the sliding window then compared with the baseline activity across all trials (movement bouts) using a right-tailed pairwise t-test. An activity rise was defined as $p < 0.05$ in consecutive 5 steps, the center of the first sliding window was used as the activity rise time.

### Statistics and reproducibility

For the fluorescent images as representative panels in the figures, no less than 3 mice from the same treatment group were checked for the validity. Data were analyzed by Prism 9 software (Graphpad) and custom code written in MATLAB (MathWorks). All statistical details for each experiment can be found in the corresponding figure legends. Data were presented as means and standard errors of the mean (SEM). We assessed the statistical significance using parametric t-test, one and two-way analysis of variance (ANOVA). Significance was considered for test statistics with a (*) $p$-value of <0.05, (**) $p$-value of <0.01, (***) $p$-value of <0.001, (****) $p$-value of <0.0001.

### Reporting summary

Further information on research design is available in the Nature Portfolio Reporting Summary linked to this article.

## Data availability

The accession number of the striatal tissue RNA-seq data is PRJNA870469. The accession number of SNc RNA-seq data is PRJNA775656. All data generated or analyzed during this study are included in this published article. Source data are provided with this paper.

## Code availability

The code used to process the fiber photometry data and optogenetics behavior analysis in this study has been deposited in GitHub [https://github.com/jdNIH/Kremen1_2025] and can also be accessed on Zendo [https://zenodo.org/records/14796086].

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

## Acknowledgements

This work is supported in part by the Intramural Research Programs of National Institute on Aging, NIH (HC, ZIA AG000944, AG000928), National Institute of Mental Health (CRG, ZIA MH002497-34), and National Nature Science Foundation of China (WDL, 32220103006). We thank Dr. Yulong Li of Peking University for providing the GRAB$_{rDA3m}$ sensor, NIMH rodent behavioral core for assisting in behavioral tests, Dr. Sarah Hawes for sharing *Calb1*$^{IRES-Cre}$ mice for the optogenetic study, Drs. Guohong Cui and Jingheng Zhou of National Institute of Environmental Health Sciences and members of Cai lab for their suggestions and technical assistance, and Drs. Da-Ting Lin of National Institute of Drug Abuse, Yangfeng Zhang of University of Exeter, and Jun Wang of Texas A & M University for their constructive feedback.

## Author contributions

H.C. conceived and designed experiments, wrote the manuscript, and prepared the figures with inputs from all authors. J.D. designed and performed stereotactic surgery, optogenetics, fiber photometry, histology, behavioral experiments and data analyses, prepared figures, wrote methods and figure legends, and wrote/edited the manuscript. L.W. designed and conducted fiber photometry experiments with head-fixed mice, performed data analyses, prepared figures and figure legends, wrote methods, and wrote/edited the manuscript. B.S. performed RNA in situ hybridization and data analyses, histology, mouse breeding, and behavioral experiments. L.S. contributed to the laser capture microdissection and RNA sequencing of the patch and matrix compartments, built the pAAV-FLEX-SaCas9-U6-sg*Gabbr1* vector, *and* wrote method for these experiments. V.M. contributed to RNAscope image analyses and histology. J.H.D. contributed to RNA sequencing data analyses. L.C. performed stereotactic surgery. W.L. contributed to data interpretation and edited the manuscript. C.R.G. provided *A2a*$^{Flp}$ mice, contributed to data interpretation, and edited the manuscript. All authors read and approved the final manuscript.

## Funding

## Competing interests

The authors declare no competing interests.
