## [Transparent Peer Review file · Nature Communications]

Molecularly Distinct Striatonigral Neuron Subtypes Differentially Regulate Locomotion

Corresponding Author: Dr Huaibin Cai

Version 0:

Reviewer comments:

Reviewer #2

(Remarks to the Author)

The manuscript submitted by Dong et al. has been extensively revised, and all comments raised by the four reviewers have been thoroughly addressed. I congratulate the authors on their comprehensive revision and fully support the publication of this excellent work in Nature Communications. I kindly request that the editors ensure the information provided in the reporting summary is fully readable. The PDF format I downloaded had truncated sentences.

Reviewer #3

(Remarks to the Author)

Overall, I enjoyed this manuscript and found it much improved in the revision. These findings will be a valuable addition to the literature.

I have one remaining minor comment/suggestion:

In Figure 1g-i, there is a circular definition – clusters of Kremen1+ SPNs define patches, and then it is stated as a finding that most Kremen1+ SPNs are in patches. A quantification of the patch distribution based on another known patch marker such as MOR1 (as done in Fig 2) would perhaps be more informative. Nevertheless, given the provided definition in the text, readers could judge for themselves. Other data provided give a clear picture overall of the distribution of Kremen1+ SPNs, which is helpful, so I would leave it to the editors and authors to decide how to approach this critique.

Reviewer #4

(Remarks to the Author)

The authors have done an excellent job addressing my concerns.

REVIEWERS' COMMENTS

Reviewer #2 (Remarks to the Author):

The manuscript submitted by Dong et al. has been extensively revised, and all comments raised by the four reviewers have been thoroughly addressed. I congratulate the authors on their comprehensive revision and fully support the publication of this excellent work in Nature Communications. I kindly request that the editors ensure the information provided in the reporting summary is fully readable. The PDF format I downloaded had truncated sentences.

Response: We appreciated the reviewer's endorsement and will work closely with the editor to address any issue during the publication process.

Reviewer #3 (Remarks to the Author):

Overall, I enjoyed this manuscript and found it much improved in the revision. These findings will be a valuable addition to the literature.

Response: We appreciated the reviewer's endorsement for the publication.

I have one remaining minor comment/suggestion:

*In Figure 1g-i, there is a circular definition – clusters of *Kremen1*⁺ SPNs define patches, and then it is stated as a finding that most *Kremen1*⁺ SPNs are in patches. A quantification of the patch distribution based on another known patch marker such as *MOR1* (as done in Fig 2) would perhaps be more informative. Nevertheless, given the provided definition in the text, readers could judge for themselves. Other data provided give a clear picture overall of the distribution of *Kremen1*⁺ SPNs, which is helpful, so I would leave it to the editors and authors to decide how to approach this critique.*

Response: As suggested by the reviewer, we have conducted additional counting of *Kremen1*⁺ neurons in the patch compartment of *Kremen1*^{2A-Cre};Ai14 mice, with the patch identified by *MOR1* immunostaining as suggested. Consistent with our RNAscope analyses (**Fig. 1i**), the number of *Kremen1*⁺ SPNs is comparable between the patch and matrix compartments. We have summarized these findings in **Supplementary Fig. 3**.

Reviewer #4 (Remarks to the Author):

The authors have done an excellent job addressing my concerns.

Response: We appreciated the reviewer's endorsement for the publication.